# Inferring transmission heterogeneity using virus genealogies: Estimation and targeted prevention

**Yunjun Zhang**[1,2]*, **Thomas Leitner**[3], **Jan Albert**[4,5], **Tom Britton**[2]

**1** Department of Biostatistics, School of Public Health, Peking University, Beijing, China, **2** Department of Mathematics, Stockholm University, Stockholm, Sweden, **3** Theoretical Biology and Biophysics Group, Los Alamos National Laboratory, Los Alamos, New Mexico, United States of America, **4** Department of Microbiology, Tumor and Cell Biology, Karolinska Institute, Stockholm, Sweden, **5** Department of Clinical Microbiology, Karolinska University Hospital, Stockholm, Sweden

* yunjun.zhang@pku.edu.cn

**Data Availability Statement:** We used three published sequence datasets: ref 28 Esbjornsson et al. (JQ698667–JQ698874), ref 29 Skar et al. (EU010264-EU010360) and ref 30 Skar et al (GU222921 to GU223066). All sequences are

## Abstract

Spread of HIV typically involves uneven transmission patterns where some individuals spread to a large number of individuals while others to only a few or none. Such transmission heterogeneity can impact how fast and how much an epidemic spreads. Further, more efficient interventions may be achieved by taking such transmission heterogeneity into account. To address these issues, we developed two phylogenetic methods based on virus sequence data: 1) to generally detect if significant transmission heterogeneity is present, and 2) to pinpoint where in a phylogeny high-level spread is occurring. We derive inference procedures to estimate model parameters, including the amount of transmission heterogeneity, in a sampled epidemic. We show that it is possible to detect transmission heterogeneity under a wide range of simulated situations, including incomplete sampling, varying levels of heterogeneity, and including within-host genetic diversity. When evaluating real HIV-1 data from different epidemic scenarios, we found a lower level of transmission heterogeneity in slowly spreading situations and a higher level of heterogeneity in data that included a rapid outbreak, while $R_0$ and Sackin's index (overall tree shape statistic) were similar in the two scenarios, suggesting that our new method is able to detect transmission heterogeneity in real data. We then show by simulations that targeted prevention, where we pinpoint high-level spread using a coalescence measurement, is efficient when sequence data are collected in an ongoing surveillance system. Such phylogeny-guided prevention is efficient under both single-step contact tracing as well as iterative contact tracing as compared to random intervention.

## Author summary

Detecting and preventing pathogen outbreaks in the background of steady and slow spread is difficult, yet highly desirable, because such transmission heterogeneity can be a main driver of an epidemic. Hence, detection of transmission heterogeneity may direct

publicly available in Genbank under the accession numbers provided.

**Funding:** Y.J.Z, J.A, and T.B acknowledge support from the Swedish research council (grant number 340-2013-5003). T.L is supported by National Institute of Allergy and Infectious Diseases/ National Institutes of Health (NIAID/NIH) under award number R01AI087520. The funders had no role in study design, data collection and analysis, decision to publish, or preparation of the manuscript.

**Competing interests:** The authors have declared that no competing interests exist.

prevention efforts and reduce future infections. While incidence and prevalence estimates may give overall indications of an epidemic's progression, they typically cannot indicate episodic outbreaks or rapid spreads in subpopulations. Likewise, detailed and reliable information about dynamic social networks is rare and not generalizable to detect local outbreaks. HIV sequence data can be used to reconstruct HIV phylogenies, which due to HIV's high evolutionary rate contain information about both transmission networks and rates of spread. Here, we use HIV phylogenies to first design a general heterogeneity detection method that can signal that there is high-level spreading present. Secondly, we develop a phylogenetic method to pinpoint which individuals that may have been infected by a super-spreader or have been involved in an outbreak. We show that using such phylogeny-guided information to prevent future HIV spread is highly efficient under many epidemiological situations, especially in typical public health situations where samples are collected through time.

## Introduction

Allocation of prevention resources to where they are needed most is important for effective disease control. Thus, identifying where disease spread has its highest intensity would allow for efficient resource allocation. The intensity of the spread is typically uneven in time and space, causing episodic and local outbreaks. One type of spread-heterogeneity comes from the situation when some individuals spread at a much higher rate than others, causing super-spreading [1], and invoking the 20-80 rule, where 20% of infected persons transmit to 80% of new infecteds [2, 3]. Such transmission heterogeneity can have impact on how fast epidemics spread, and may affect the outcome of control efforts [4].

While epidemiological methods using incidence data can pick up the signal of increased number of cases, which may indicate an outbreak or be the result of increased surveillance, they can not identify individual-level transmission heterogeneity. To assess transmission heterogeneity on the individual-level transmissibility, contact tracing based on interviews, reviews of clinical records, and partner follow-up have traditionally been used. Such follow-up may be slow, expensive and inaccurate, however. Several studies have reported that interview-based information about sexual contacts where HIV-1 transmission might have taken place often was not in agreement with the phylogenetic history of the transmitted virus [5, 6]. Early studies showed that virus phylogenies reflect their underlying transmission histories [7], and more recent work has shown that phylogenies carry information about the underlying population structure [8–10] and the degree distributions of the sexual contact network [11–13] both of which contribute to the transmission heterogeneity.

The Multi-state birth-death (MSBD) models have been used to model population structure and infer the related transmission heterogeneity [8, 14]. The rationale of these models is to associate each cluster (or subpopulation) to a state in the MSBD model, with clusters differing in their transmission dynamics through time. An important limitation of the MSBD models is either requiring the correspondence between the tips and the states is known in advance [15] or using fixed positions of state changes [10]. An efficient method of model-based genetic clustering has recently been proposed in [9], which is based on fitting a Markov-modulated Poisson process representing the evolution of transmission rates along the tree relating different infections. Though it performs well in clustering lineages of different risky groups, the method is not designed to infer basic reproduction number which is also of certain interest in epidemiloigcal practice.

In addition, because both incidence and pathogen phylogenies carry information about the epidemic, several efforts have been made to combine these data into comprehensive heterogeneity inference [16–18]. Interestingly, Li et al [19] recently showed that quantification of transmission heterogeneity was more accurately estimated using the pathogen phylogeny alone rather than in combination with incidence data. They describe the transmission heterogeneity as the variation among the offspring distributions, i.e., the number of secondary cases caused by each infected individual. The offspring distribution, however, is not only affected by the individual-level transmissibility but also by the length of the infectious period of each infected individual.

In this study, we propose a heterogeneous birth-death model to describe the variation of the individual-level transmissibility, with birth rates corresponding to transmissibility (or transmission rates) and death rates to diagnosis rates (or removal rates). We model the transmission heterogeneity by letting each individual draw an independent random transmission rate from a continuous distribution, so the transmission heterogeneity is captured in a parameter quantifying the amount of variation in this distribution. Transmission heterogeneity may exist due to varying degrees of social activity as wells varying transmission risk upon contact, e.g., due to varying viral load in different disease stages. We then develop a general inference procedure to estimate model parameters and to test whether or not there is significant transmission heterogeneity, and thereafter we develop a method that identifies evolutionary lineages that likely have been involved in many transmission events of the epidemic. We use the second method to direct targeted prevention efforts, by means of (additional) contact tracing, towards individuals that have been involved in elevated spread as compared to random intervention. To address the realistic situation where not all persons have been sampled in an epidemic, we investigate the effects of sampling fractions, situations where samples are taken cross-sectionally and when samples are gathered in an ongoing surveillance system. We investigate the effects of within-host diversity in simulations on both the general heterogeneity test as well as the targeted prevention. We also test our estimate of transmission heterogeneity in facing of realistic complexities such as autocorrelation in transmission rates and varying transmission rate through time. We apply the heterogeneity model and estimate parameters on 3 real datasets sampled from different HIV-1 epidemics.

## Materials and methods

### A transmission model with heterogenous transmissibility

In order to estimate transmission heterogeneity in a population and, further, the effect of targeted prevention on individuals associated with high transmissibility, we develop a new transmission model, referred to as the heterogeneous birth-death (HBD) model, in which individuals in a community may have different rates of infecting new individuals. The difference in transmissibility could be caused by variable virus load leading to variable infection probability upon contact, and/or variable social contact rate. Here, we refer to any combination of these concepts as transmissibility.

Specifically, in our model each infected individual $i$ gets assigned a transmissibility rate $\lambda_i$ by randomly drawing the rate from a *Gamma* distribution with mean $\mu_\lambda$ and standard deviation $\sigma_\lambda$. The gamma distribution is a common choice when modeling individual heterogeneity for a parameter. While infectious, this individual infects new individuals randomly in time, with rate $\lambda_i$ per unit of time. Also we assume that the transmission rates of the infector and the infectee in a transmission event are independent. Later we will evaluate our model under the condition with autocorrelation of transmission rates.

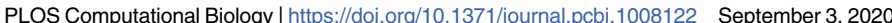

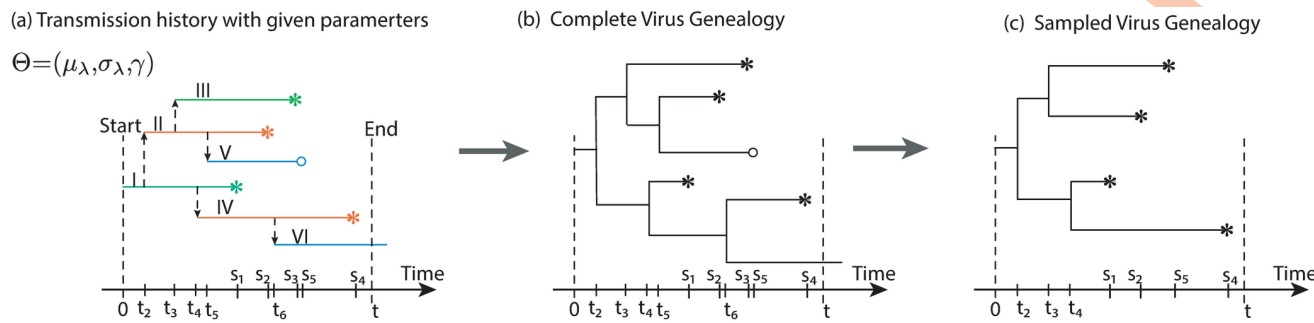

**Fig 1. The connection between transmission history $\mathcal{H}$ (a), complete genealogy $\mathcal{G}$ (b), and sampled genealogy $\mathcal{G}^{(S)}$ (c).** In the model, a transmission history (a) is simulated with the heterogeneous birth-death transmission model under a given set of parameters ($\Theta$). The *i*-th ($i = I, \cdots, V$ in Roman numeral notation) individual was infected and stopped infection at $t_i$ and $s_i$ ($i = 1, \cdots, 5$ in Arabic numerals) respectively. There are three categories of individuals at the end of the simulation: sampled individuals, i.e. diagnosed and sequenced (tips labeled with $-^*$), diagnosed but unsampled individuals (tips labeled with $-\circ$), and undiagnosed individual (unlabeled tips). A complete virus genealogy $\mathcal{G}$ (b) contains the genealogical lineages from all the infected individuals. In the case of no within host diversity it coincides with the transmission tree $\mathcal{T}$ being the transmission history $\mathcal{H}$ but without the direction of transmission. The sampled genealogy $\mathcal{G}^{(S)}$ (c) is derived from the complete genealogy by removing edges from unsampled individuals.

Infected individuals remain infectious until they are diagnosed—it is then assumed that individuals are immediately treated such that they no longer are infectious and instead immune [20]. The time to diagnosis, or more precisely to successful treatment, is assumed to be exponentially distributed with rate $\gamma$ common for all individuals (hence random and with no systematic heterogeneity). The diagnosed individuals are sampled for virus sequencing with probability $\rho_{SD}$ (referred to as the *sequencing ratio*). Consequently, there are three categories of infected people: a) the undiagnosed (who continue to spread the disease), b) the diagnosed/treated who have not been sampled for sequencing, and c) the sequenced diagnosed/treated individuals. The ratio of the sequenced diagnosed over all the infected (i.e. c/(a + b + c)) is referred to as the *sampling ratio* in this study. In contrast with the sequencing ratio (i.e. c/(b + c)) which is usually assumed to be known in advance, the sampling ratio is unknown in the analysis of virus sequence data.

In our model, a transmission history is generated with our heterogeneous birth-death transmission model under a given set of parameters ($\Theta = (\mu_\lambda, \sigma_\lambda, \gamma)$). The transmission history (denoted as $\mathcal{H}$) contains all information about who-infected-whom in calendar time (Fig 1). The corresponding tree without information about who-infected-whom is called the transmission tree and denoted as $\mathcal{T}$. In general, the transmission tree cannot be observed from virus sequence data. However, what we can infer is the tree that describes how the virus sequences from all the infected individuals are related, i.e. the virus genealogy $\mathcal{G}$. If there is no within-host virus diversity, then the virus genealogy and transmission trees are identical: $\mathcal{G} = \mathcal{T}$. If within-host diversity is taken into account (while still assuming only one strain is transferred at transmission), then the coalescence times of the sampled virus sequences may coalesce further back in time than the time of infection, with the effect that branch lengths and even tree topologies may differ from the underlying transmission tree [21]. The estimation method in this study was first developed under the former situation (i.e. without within-host diversity), and was extended to the latter case in the simulation.

The input to our analysis is the pathogen (virus) genealogy (denoted as $\mathcal{G}^{(S)}$), i.e. the time-scaled phylogeny that can be inferred from the viral DNA sequences from sampled patients (i.e. the diagnosed and sequenced individuals) in an epidemic spanning from time 0 to *t*. The sampled virus genealogy $\mathcal{G}^{(S)}$ is obtained by removing the unobserved (undiagnosed and diagnosed but not sampled) individuals and the corresponding coalescent events from $\mathcal{G}$ (Fig 1).

The parameters to be estimated in analysis are the mean $\mu_\lambda$ and standard deviation $\sigma_\lambda$ of the transmissibility and the mean time to diagnosis/treatment $\gamma^{-1}$, with the main focus on the degree of heterogeneity of transmissibility, either measured in absolute terms as $\sigma_\lambda$ or in relative terms as the coefficient of variation $CV_\lambda = \sigma_\lambda/\mu_\lambda$. The $CV$ is a dimensionless quantity where standard deviation, as a measure of dispersion, is normalized by the mean that facilitates comparisons across data sets. When either $\sigma_\lambda$ or $CV_\lambda$ are large, then that implies that a few individuals will tend to infect at a much higher rate than others. We also infer the basic reproduction number $R_0$, which is the ratio of the mean transmissibility rate and the diagnosis rate, i.e. $R_0 = \mu_\lambda/\gamma$, being the average number of infections caused by an infected before diagnosis and treatment.

Our transmission model is an extension of the homogeneous birth-death model in [22], which does not include transmission heterogeneity. Moreover, Leventhal et al [23] explicitly take the finite population into account by considering depletion of susceptible. Here, we neglect this depletion but investigate this simplification in the sensitivity analysis. Recently, Li et al [19] studied a related model where heterogeneity in the number of infections was described in a generation sense; while they assumed that all infections happen at the end of the infectious period, in our model infectious individuals infect new individuals at rate $\lambda_i$ *during* the infectious period. On the other hand, their model also allow for the possibility to have a highly concentrated distribution of number of infections, for example caused by a social network with most individuals having very similar number of contacts.

## Inferring transmission heterogeneity from sampled viral genealogy

There are two difficulties of using the sampled genealogy $\mathcal{G}^{(S)}$ to infer transmission heterogeneity: one is the lack of transmission direction (who-infected-whom) in $\mathcal{G}^{(S)}$, and the other that $\mathcal{G}^{(S)}$ has incomplete transmission chains [19]. We deal with these two difficulties by using a coordinate-ascent algorithm to estimate both the parameters and the direction of transmission in a genealogy, and, additionally, a data imputation correction to handle the incomplete transmission history. We will first introduce the coordinate-ascent algorithm and consequently give a description about the analysis of the sampled genealogy $\mathcal{G}^{(S)}$.

**Inference with coordinate-ascent algorithm.** The coordinate-ascent (CA) algorithm works with a complete genealogy as input, that is, the algorithm assumes there is no within-host diversity and that we observe and sequence all infected individuals up to a certain time $t$. Later, these assumptions have been relaxed to analyze the sampled genealogy.

The CA algorithm has been developed on the basis of the likelihood function for a transmission history $\mathcal{H}$. Under the assumptions of our HBD model, not only the transmission events caused by different individuals occur independently, but also the multiple transmission events caused by one individual are independent. So the likelihood for the whole transmission history can be evaluated in terms of individuals. Let $x_i$ and $d_i$ $(i = 1, \cdots, n)$ denote the number of infections and the duration of the infectious period for individual $i$, information which is contained in the transmission history $\mathcal{H} = \{(d_i, x_i)\}_{i=1}^n$. The likelihood function is then given by

$$
\begin{aligned}
L(k, \theta, \gamma | \mathcal{H}) \\
\propto \prod_{i=1}^n Prob(d_i, x_i | k, \theta, \gamma) \\
= \prod_{i=1}^n \int \lambda_i^{x_i} e^{-\lambda_i d_i} * \gamma^{\mathbb{1}_{DG}(i)} e^{-\gamma d_i} * \frac{1}{\Gamma(k)\theta^k} \lambda_i^{k-1} e^{-\frac{\lambda_i}{\theta}} d\lambda_i \\
= \prod_{i=1}^n \frac{\Gamma(x_i + k)}{\Gamma(k)} \frac{\theta^{x_i}}{(1 + \theta d_i)^{x_i+k}} * \gamma^{\mathbb{1}_{DG}(i)} e^{-\gamma d_i}
\end{aligned}
\tag{1}
$$

where $\Gamma(\cdot)$ is the gamma function, and the indicator $\mathbb{1}_{DG}(i)$ in (1) indicates if the infectious period $d_i$ has ended before the end of the observation period. The parameters of $(k, \theta)$ are parameterizations of *Gamma* distribution under which the mean and standard deviation can be represented as $\mu_\lambda = k\theta$ and $\sigma_\lambda = \sqrt{k}\theta$ respectively.

Consequently, the maximum likelihood estimates of the model parameters are obtained by equating the gradient of the log-likelihood function to zero. Particularly, the estimates of $\hat{k}$ and $\hat{\theta}$ are obtained by solving the following equations:

$$\hat{k} = \frac{\sum_i \frac{x_i}{1+\hat{\theta}d_i}}{\sum_i \frac{\hat{\theta}d_i}{1+\hat{\theta}d_i}}, \text{ and } \sum_i \left(\sum_{j=1}^{x_i} \frac{1}{\hat{k}+j} - \ln\left(1 + \hat{\theta}d_i\right)\right) = 0. \tag{2}$$

Moreover, the estimate of $\hat{\gamma}$ is

$$\hat{\gamma} = \frac{\sum_i \mathbb{1}_{DG}(i)}{\sum_i d_i}, \tag{3}$$

which is simply the observed number of recoveries divided by the total duration of the infectious periods in the transmission history $\mathcal{H}$.

When we observe the virus genealogy $\mathcal{G}$ rather than the transmission history $\mathcal{H}$, we can no longer estimate the parameters as described above since $\mathcal{G}$ does not contain information of transmission direction, i.e. who infected whom. In other words, we can then not determine $(d_i, x_i)_{i=1}^n$, so the estimates in (2) are not available. However, the estimator $\hat{\gamma}$ based on (3) is still available without the information of transmission directions: even if each infectious period cannot be determined, the sum of infectious periods $\Sigma_i d_i$ is independent of transmission directions and equals the sum of branch lengths in $\mathcal{G}$ (denoted as $|\mathcal{G}|$). The estimator $\hat{\gamma}$ can hence be calculated before the CA algorithm is introduced below, and it is given by:

$$\hat{\gamma} = \frac{\#(\mathcal{G})}{|\mathcal{G}|} \tag{4}$$

where $\#(\mathcal{G})$ is the number of diagnosis events in $\mathcal{G}$.

In the following, we focus on the estimation of parameter $k$ and $\theta$. We treat the transmission directions of the branching events in $\mathcal{G}$ as latent variables, and propose the CA algorithm which alternates between estimating the model parameters ($\Theta = \{k, \theta\}$) and reconstructing the transmission history $\mathcal{H}$. Here the reconstruction is implemented in a form of labeling the branches of $\mathcal{G}$ (as illustrated in Fig 2). The branching events following the labeled branches are the transmission events caused by the individuals with the corresponding labels. The CA algorithm traverse the genealogy $\mathcal{G}$ *slice-by-slice* from the leaves up to the root (Fig 2). Each slice (illustrated in Fig 2(a)) consists of branches with the same *height*, defined as the number of edges on the longest downward path from this branch to a tip (i.e. similar to the definition of the height of a node). For example, the external branches have the height of 0, so they consist of a slice. The branches within a particular slice come from different infected individuals, and the transmission event following these branches are hence independent. Therefore, labeling these branches can be performed in parallel.

The inference process begins with the external branches which are assigned to different individuals, yielding the initial estimate of the transmission history $\hat{\mathcal{H}}^0$. Because of the independence assumption in the HBD model, the likelihood function in Eq (1) is also valid for the partially reconstructed transmission history $\hat{\mathcal{H}}^0$, and the initial estimate of $\hat{\Theta}^0$ is hence obtained via (2). Further, for all unlabeled branches in a slice, the CA algorithm alternates between updating transmission history (i.e. labeling the branches based on the present

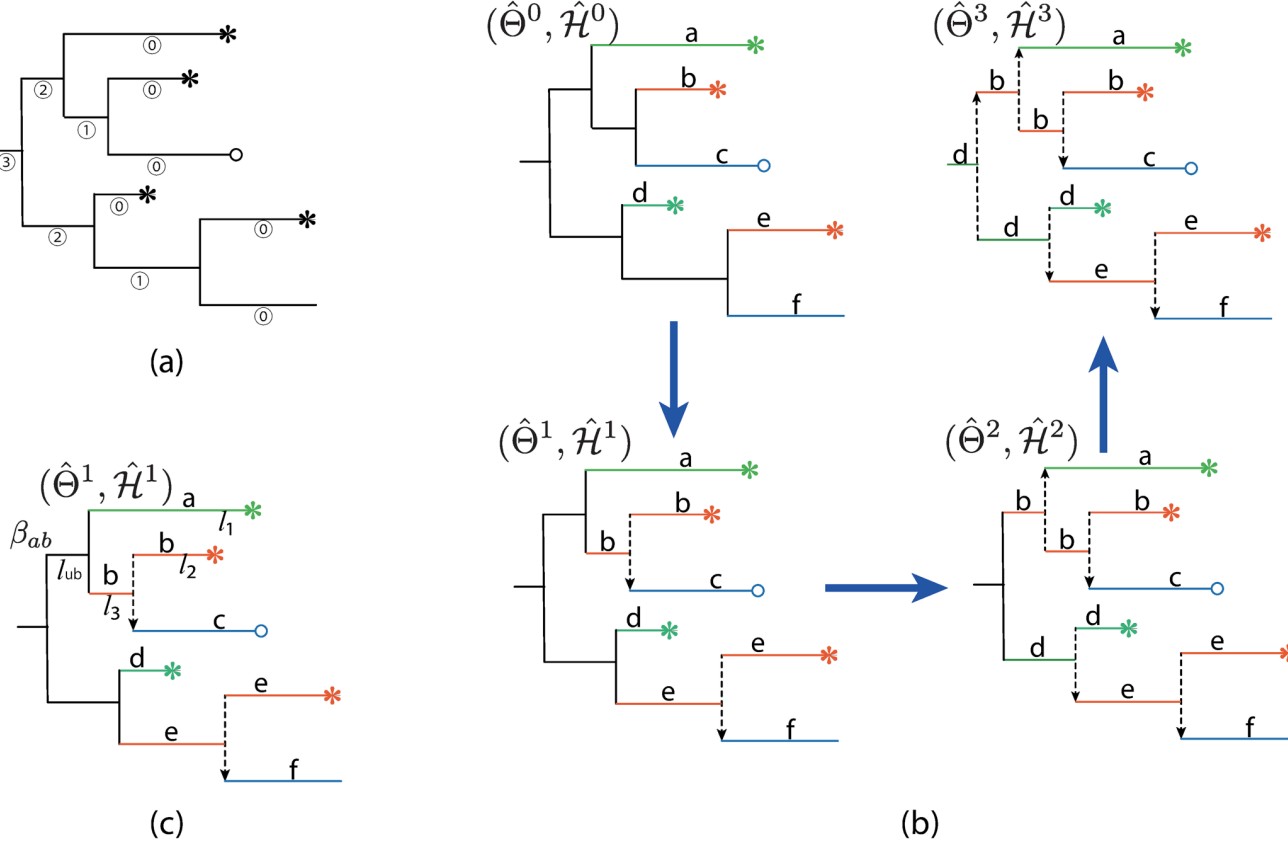

**Fig 2. Schematic representation of the coordinate-ascent algorithm.** (a) The branches in the genealogy $\mathcal{G}$ have been divided into 4 slices (labeled as ⓪ to ③). (b) The algorithm performs estimation of model parameters (denoted as $\Theta$) and reconstructing the transmission history $\mathcal{H}$ (or labeling the branches (with different letters) of a genealogy) simultaneously under the likelihood framework. The algorithm starts from the external branches (labeled as ⓪) which come from different individuals, and proceeds backwards slice by slice. The internal branches with the same label represent the infectious period of a particular individual, who is the infector of the transmission events during that period. (c) Given the present estimate of model parameters and transmission history $(\hat{\Theta}^1,\hat{\mathcal{H}}^1)$, the transmission histories of individual $a$ and $b$ are $(d_a = l_1, e_a = 0)$ and $(d_b = l_2 + l_3, e_b = 1)$ respectively. For the unlabeled branch $l_{ub}$, its labeling coefficient is evaluated as $\beta_{ab} = \frac{Prob\{(l_1+l_{ub},1),(l_2+l_3,1)|\hat{\Theta}^1\}}{Prob\{(l_1,0),(l_2+l_3+l_{ub},2)|\hat{\Theta}^1\}}$.

estimate of model parameters) and updating the estimation of model parameter based on the updated transmission history, until convergence.

More precisely, we evaluate the *labeling coefficient* for an unlabeled branch as the ratio of the probabilities of labeling the branch to its left and right descendant branches. Suppose that the unlabeled branch has a length of $l_u$, and that its left and right descendant branches were already labeled as $a$ and $b$ respectively, that is, with reconstructed transmission history of $(d_a, e_a)$ and $(d_b, e_b)$, respectively. Labeling the new branch as $a$ or $b$ stands for updating the corresponding transmission history by prolonging for another $l_u$ time unit and adding one more transmission event. Here the *labeling coefficient* $\beta_{ab}$ is calculated as (illustrated in Fig 2(c)):

$$\beta_{ab} = \frac{Prob\{(d_a + l_u, e_a + 1), (d_b, e_b)|\hat{\theta}, \hat{k}, \hat{\gamma}\}}{Prob\{(d_a, e_a), (d_b + l_u, e_b + 1)|\hat{\theta}, \hat{k}, \hat{\gamma}\}}. \tag{5}$$

where the probabilities such as $Prob\{(d_a + l_u, e_a + 1), (d_b, e_b)|\hat{\theta}, \hat{k}, \hat{\gamma}\}$ is evaluated via (1) by plugging in these present estimates of $\hat{\theta}, \hat{k}$, and $\hat{\gamma}$. Consequently, if $\beta_{ab} > 1$, then the new

branch is labeled as its left descendant branch (*a* in this example); otherwise, as its right descendant branch (*b* in this example).

**Analysis of sampled virus genealogy.** In this part, we describe the analysis of the sampled genealogy $\mathcal{G}^{(S)}$ based on the CA algorithm. Since $\mathcal{G}^{(S)}$ suffers from partial sampling, the CA algorithm cannot be applied directly. For a given tuning parameter $p$ ($0 < p < 1$, often slightly smaller than 1), we focus on the part of $\mathcal{G}^{(S)}$ up to time $t - L_p$ (denoted as $\mathcal{G}^{(S)}_{t-L_p}$), where $L_p$ is the $p \times 100$ quantile of the lengths of the external branches of $\mathcal{G}^{(S)}$. We set $p$ to a high value ($p = 0.9, 0.85,$ and $0.8$) to get a local genealogy $\mathcal{G}^{(S)}_{t-L_p}$ which contains a relative large part of information about the epidemic dynamics up to time $t - L_p$. Specifically, our analysis consists of three steps: firstly, we estimate the sampling ratio (i.e. the number of the diagnosed and sequenced divided by all the infected) of the local genealogy $\mathcal{G}^{(S)}_{t-L_p}$; secondly, we apply the CA algorithm to the local genealogy $\mathcal{G}^{(S)}_{t-L_p}$ by assuming that all individuals infected within $[0, t - L_p]$ are sampled; thirdly, we correct these parameter estimates by taking advantage of the estimated sampling ratio in the first step. Also, the results under different values of $p$ are averaged out to generate the final estimation.

Firstly, the sampling ratio of the local genealogy $\mathcal{G}^{(S)}_{t-L_p}$, or in other words, the number of the unsampled individuals up to time $t - L_p$, has been estimated. As defined above $L_p$ is the $p \times 100$ percentile of the lengths of the external branches of $\mathcal{G}^{(S)}$, so $L_p$ provides a conservative estimate of the $p \times 100$ percentile for the distribution of infectious periods. Hence, we assume that a fraction $p$ of the individuals infected within $[0, t - L_p]$ will be diagnosed before time $t$. In addition, a fraction $\rho_{SD}$ of the diagnosed individuals was selected to do sequencing (i.e. the sequencing ratio is $\rho_{SD}$), so the sampling ratio of $\mathcal{G}^{(S)}_{t-L_p}$ is $\rho_{SD} * p$. Denoting $N(\mathcal{G}^{(S)}_{t-L_p})$ as the number of individuals collected in the local genealogy $\mathcal{G}^{(S)}_{t-L_p}$, so there are $N(\mathcal{G}^{(S)}_{t-L_p})\left(\frac{1}{\rho_{SD}*p} - 1\right)$ individuals which infected before time $t - L_p$ but were not sampled.

Next, we apply the CA algorithm to the local genealogy $\mathcal{G}^{(S)}_{t-L_p}$ and obtain the raw estimates of the mean (denoted as $\hat{\mu}^0_\lambda$) and the standard deviation (denoted as $\hat{\sigma}^0_\lambda$) of the transmissibility rate. Furthermore, the raw estimate of the transmission heterogeneity is $\hat{CV}^0 = \hat{\sigma}^0_\lambda / \hat{\mu}^0_\lambda$.

Last, we study the correction of the raw estimates by considering the unsampled individuals. This correction is performed on the basis of data imputation. For the sampled individuals, their estimated transmissibility rates shall be upscaled based on the sampling ratio by allowing for partial sampling. For the unsampled individuals, we assume that they have the same infectious period and the same transmissibility rate because of no further information about their difference. The corrected values for the sampled individuals together with the substituted values for the unsampled individuals are combined to yield the corrected estimation of model parameters as follows (Please see S1 Text for details about the derivation of (6)-(8)):

The corrected estimation of the diagnosis rate is

$$
\hat{\gamma}^p = \frac{\#(\mathcal{G}^{(S)}_{t-L_p})/\rho_{SD}}{|\mathcal{G}^{(S)}_{t-L_p}| + N(\mathcal{G}^{(S)}_{t-L_p})(\frac{1}{\rho_{SD} * p} - 1) * \bar{l}_{LE}}, \tag{6}
$$

where $\bar{l}_{LE}$ is the mean length of the external branches in $\mathcal{G}^{(S)}$, and is used as the infectious period of the unsampled individuals.

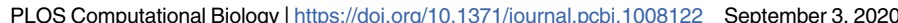

In addition, the corrected estimates of the mean and standard deviation of transmissibility rate are

$$\hat{\mu}_\lambda^p = \hat{\mu}_\lambda^0 + (1 - \rho_{SD} * p)\lambda_{us}, \tag{7}$$

and

$$\hat{\sigma}_\lambda^p = \sqrt{\frac{(\hat{\sigma}_\lambda^0)^2}{\rho_{SD} * p} + \frac{1 - \rho_{SD} * p}{\rho_{SD} * p}(\hat{\mu}_\lambda^0 - \rho_{SD} * p * \lambda_{us})^2} \ , \tag{8}$$

where $\lambda_{us}$ is the substituted transmissibility rate for the unsampled individuals. Furthermore, the corrected estimation of transmission heterogeneity is $\hat{CV}^p = \hat{\sigma}_\lambda^p / \hat{\mu}_\lambda^p$.

The value of $\lambda_{us}$ shall vary with the level of heterogeneity. Under the homogeneous situation (i.e. $CV_\lambda = 0$), all the infected individuals have the same transmissibility rates, so we set the transmissibility rates of the unsampled individuals to be the same as the average rate of the sampled individuals, that is, $\lambda_{us} = \hat{\mu}_\lambda^0$. On the other hand, under the situation with extremely high level of heterogeneity, a small proportion of individuals have very high level of transmissibility rates, while a large proportion of individuals have small level of transmissibility rates. The former will cause most of the infections, while the latter will cause few or no infections. Note that the transmission events caused by an individual could be reconstructed by the information from its descendants. The sampled individuals in the genealogy $\mathcal{G}^{(S)}$ are more likely to be the descendants of the highly infectious individuals. Hence the coalescent events in the genealogy $\mathcal{G}^{(S)}$ are more likely corresponding to the transmission events caused by these highly infectious individuals, even if some of them are not sampled while their descendants are sampled instead. Hence the transmissibility rates estimated from the sampled genealogy reflect the rates with which highly infectious individuals spread the disease. Based on this consideration, under the situation with high level of heterogeneity, we assume that unsampled individuals have low transmissibility rates and that they cause few or no infections before being diagnosed. In other words, we set the constant transmissibility rate $\lambda_{us}$ for the unsampled individuals to $\lambda_{us} = \hat{\gamma}$, where $\hat{\gamma}$ is the estimated recovery rate. For the general case we compromise between the two extreme situations on the basis of the level of heterogeneity as follows:

$$\lambda_{us} = \hat{\mu}_\lambda^0 + \frac{1 - \exp(CV^0)}{1 + \exp(CV^0)}(\hat{\gamma} - \hat{\mu}_\lambda^0), \tag{9}$$

where $\hat{CV}^0$ is a raw estimation about the level of heterogeneity as defined above.

## Targeted prevention based on phylogenetic information

An important application for heterogeneity detection is to optimize the epidemic control policy when transmission heterogeneity exists. As pointed out in [1], focusing on the highly infectious individuals will greatly increase the efficiency of disease control policy. To facilitate such an application, we propose a phylogenetic index—the Number of Coalescent Events (NCE) which helps to identify potentially highly infectious individuals. We investigate the effectiveness of using this index with the intention of controlling the spread of disease in our heterogeneous birth-death model.

Specifically, for each tip in the sampled genealogy $\mathcal{G}^{(S)}$, we calculate the *NCE* within a fixed time-period $t_c$ prior to the time of sampling of the individual. The taxa (diagnosed individuals) with higher *NCE*-values are more likely to be phylogenetically linked with unsampled highly infectious individuals. Thus, performing contact tracing on the individuals with high *NCE*-values should identify undiagnosed and highly infectious individuals and/or their undiagnosed

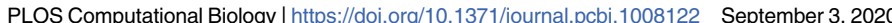

descendants, and therefore efficiently reduce further disease spread. We refer to contact tracing based on *NCE*-value as phylogeny-guided contact tracing. We evaluate different threshold values $m$, i.e. where individuals who have $NCE \geqslant m$ within $t_c$ prior to being sampled are selected for contact tracing. We also evaluate both single-step and iterative contact tracing schemes [24] to show the full beneficial effects of phylogeny-guided contact tracing policies.

## Overview of simulation study

To validate our general transmission heterogeneity detection method, we tested the performance of the inference algorithm on simulated data. As reported the average time between infection and diagnosis/treatment for HIV-1 in Sweden is around 2.5 years [11, 25, 26], so we set the rate of diagnosis at $\gamma = 1/2.5$ with year being our unit of time. We set $R_0 = 2.5$ (implying that $\mu_\lambda = 1$) which is in agreement with the previous study on the transmission of HIV in Switzerland [22]. We varied the level of transmission heterogeneity (i.e. $CV = \sigma_\lambda/\mu_\lambda$) from 0 (no heterogeneity) to 5 (extremely high heterogeneity).

Unless otherwise stated, each simulation began with one infection and was stopped when there were 100 diagnosed individuals (the average number of infected but not yet diagnosed were 130). Among these 100 diagnosed individuals, 90 diagnosed individuals were sampled for sequencing to reconstruct the viral genealogy, that is, the sequencing ratio was set to $\rho_{SD} = 0.9$. In all, the average sampling ratio in our simulations was $90/230 \approx 0.39$. We assume that there was no delay between the time of diagnosis and sampling. Please see S1 Text for details of simulation.

All the codes for simulation and inference are available at github.com/yunPKU/HeteroInfer.

## Results

### Estimation of heterogeneity and other model parameters on simulated datasets

The epidemic parameters describing the recovery rate ($\gamma$), average infectivity rate ($\mu_\lambda$), and the transmission heterogeneity ($\sigma_\lambda$ and $CV_\lambda = \sigma_\lambda/\mu_\lambda$) were estimated from our heterogeneous birth-death model and the described inference methodology (Fig 3A, 3B, 3D and 3E). While there was a slight bias trend over $CV_\lambda = \sigma_\lambda/\mu_\lambda$, the bias was well within the corresponding 95% confidence interval in each case. Hence, under a wide range of epidemic situations with different levels of heterogeneity ($CV_\lambda$) the means of the epidemic parameter estimates were close to the corresponding true values. The inference of the basic reproduction number ($R_0$), showed slight upward biases under low level of heterogeneity and slight downward biases under high level of heterogeneity (Fig 3B). However, these biases were also covered by the 95% confidence intervals.

The estimated $CV_\lambda$ showed non-negligible downward bias. Since the coefficient of variation $CV_\lambda$ is the inverse square-root of the shape parameter in the *Gamma* distribution, this observation agrees with earlier findings showing that the maximum likelihood estimate of the shape parameter tend to upward biased [27], leading to $CV_\lambda$ being underestimated.

The upward bias trend of $\hat{\gamma}$ over heterogeneity level was potentially due to our simulation process conditioning on reaching 100 tips. Given this condition, the average branch length of the simulated tree decreased with the heterogeneity level $CV_\lambda$ (as shown in S1 Fig), indicating the simulation datasets were biased in favor of 'shorter' trees under higher level of heterogeneity. Therefore, the estimate of $\hat{\gamma}$ was upward biased under high $CV_\lambda$. On the other hand, the estimate of $\hat{\mu}_\lambda$ remained relatively stable. This is due to the fact that we used $\lambda_{us}$ as the substituted

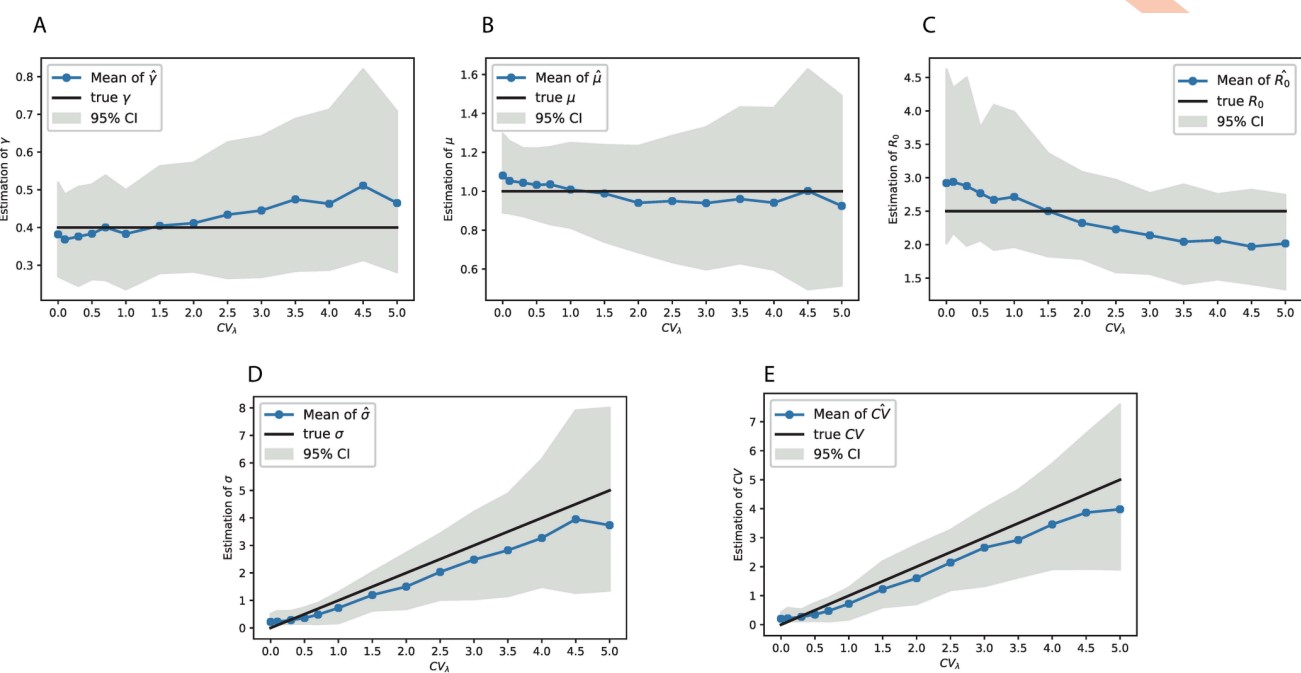

**Fig 3. Estimates of the heterogeneity and other epidemiological parameters from simulated virus genealogy.** The simulations were performed under various true levels of heterogeneity ($CV_\lambda = \sigma_\lambda / \mu_\lambda$). In each panel, the black line denotes the true value that was used to generate the simulated data, and the colored curve and the shaded area denote the mean and 95% confidence interval from 200 simulations, respectively. A estimate of recovery rate ($\gamma$), B estimate of average infectivity rate ($\mu_\lambda$), C estimate of the basic reproduction number ($R_0$), D estimate of the standard deviation of infectivity rate ($\sigma_\lambda$), and E estimate of the coefficient of variation ($CV_\lambda$).

transmissibility rate for the unsampled individuals. We explained in section of *Analysis of sampled virus genealogy* that, under the high level of heterogeneity, $\lambda_{us}$ was close to the estimated recovery rate $\hat{\gamma}$ which is considerably smaller than the true value of $\mu_\lambda$. So this correction method reduced the upward bias of $\hat{\mu}_\lambda$ under high heterogeneity situation, and hence resulted in a stable estimate. In all, the upward bias trend of $\hat{\gamma}$ together with the stable estimate of $\mu_\lambda$ over $CV_\lambda$ resulted in the downward bias trend of the estimated $\hat{R}_0$ over heterogeneity levels.

In the simulation study we also vary time scales by fixing the average infectivity rate $\mu_\lambda = 1$ and gradually increasing the average length of infectious period $\gamma^{-1}$ from 1.5 to 3, corresponding to the basic reproduction number $R_0$ increased from 1.5 to 3. As before the simulation began with one infection and was stopped when there were 100 diagnosed individuals. The results are summarized in S2 and S3 Figs.

The estimates of transmission heterogeneity ($\sigma_\lambda$ and $CV_\lambda$) were robust to differences in the average length of infectious period $\gamma^{-1}$ (S2 Fig). The effect on the estimated $CV_\lambda$ was small unless the true heterogeneity level was higher than 3 (S2A Fig). We also found that the estimated $\hat{R}_0$ remained upward biased under the scenarios with shorter infectious period ($\gamma^{-1} = 1.5$ and 2). As our simulation process being conditioning on reaching 100 tips, the simulated datasets were likely to be biased in favor of "faster" trees which resulted in the overestimated $R_0$ [28].

**Inference of heterogeneity when transmission rates are autocorrelated.** Our new method was developed under the assumption that each infected individual has an independent transmission rate. In reality, however, risk factors of transmissibility such as behavior often exhibit a high degree of homophily, such that the transmission rate of a newly infected individual may be similar to that of the donor. We therefore evaluated our method when transmission rates are autocorrelated and made a comparison with the previously developed Markov-

Modulated Poisson Process (MMPP) method [9] which was developed by exploiting the auto-correlation of transmission rates to genetic clustering as well as to estimate different transmission rates of different subpopulations.

We adopted the simulation scenario in [9] to divide the whole susceptible population into two subpopulations with different transmission rates ($\lambda_1$ and $\lambda_2$ respectively) and simulated the outbreak in two ways, that is, with and without autocorrelation of transmission rates. In the former case, each newly infected individual has the same transmission rate as its infectee with the probability $1 - \pi_s$ and switch to another transmission rate with the probability $\pi_s$. Here we set the transmission rates as $\lambda_1 = 0.9$ and $\lambda_2 = 8.1$ (9-fold faster than the other as in [9]). The switching probability is $\pi_s = 0.2$ or $\pi_s = 0.8$, corresponding to high level of autocorrelation with low heterogeneity ($CV_\lambda = 0.46$) and low level of autocorrelation with high heterogeneity ($CV_\lambda = 1.05$) respectively. The evaluation under each condition is performed based on 100 simulated replicates.

In the case without autocorrelation, each newly infected individual draws its transmission rate independently from a binary probabilistic distribution, that is, choosing $\lambda_1$ with probability $1 - \pi_c$, and choosing a larger transmission rate $\lambda_2$ with probability $\pi_c$. We set $\pi_c = 0.1$, corresponding to the proportion of risky subpopulation in [9]. We made the comparison for two settings, one with lower level of heterogeneity: $\lambda_1 = 2$ and $\lambda_2 = 6$ (corresponding to the mean transmission rate $\mu_\lambda = 2.4$ and the heterogeneity $CV_\lambda = 0.5$); and the other with higher level of heterogeneity: $\lambda_1 = 1$ and $\lambda_2 = 15$, corresponding to the same $\mu_\lambda = 2.4$ but now $CV_\lambda = 1.75$.

Under the situation of no autocorrelation of transmission rates, it is clear that the new method generates more accurate estimates of the mean ($\mu_\lambda$) and the heterogeneity ($CV_\lambda$) of two transmission rates (S9A and S9B Fig). For the low heterogeneity condition with true values of $\mu_\lambda = 2.4$ and $CV_\lambda = 0.5$ (S9A Fig), the medians of the estimates for $\mu_\lambda$ and $CV_\lambda$ are 2.32 and 0.7 from our new method and are 4.65 and 0.74 from the MMPP respectively. For the high heterogeneity condition with true values of $\mu_\lambda = 2.4$ and $CV_\lambda = 1.75$ (S9B Fig), the medians are 2.19 and 1.56 from our new method and are 6.35 and 1.04 from the MMPP respectively.

Under the situation with autocorrelation of transmission rates, the new method is comparable with the MMPP in terms of estimation accuracy (S9C and S9D Fig). For the low switching probability ($\pi_s = 0.2$) condition with true values of $\mu_\lambda = 6.49$ and $CV_\lambda = 0.46$ (S9C Fig), the medians of the estimates for $\mu_\lambda$ and $CV_\lambda$ are 5.22 and 0.79 from the new method and are 8.73 and 0.99 from the MMPP respectively. For the high switching probability $\pi_s = 0.8$ condition with true values of $\mu_\lambda = 3.18$ and $CV_\lambda = 1.05$ (S9D Fig), the medians of the estimates are 2.64 and 1.27 from our new method and are 5.75 and 1.00 from the MMPP respectively.

**Parameter estimates remain accurate under finite population sizes and biased sequencing ratio $\rho_{SD}$.** The above results show good performance of the proposed inference method under the heterogeneous birth-death model assuming that there is no depletion of susceptibles (corresponding to an infinite population) and that the sequencing ratio $\rho_{SD}$ was known (but the sampling ratio is unknown). In real applications, however, these two assumptions do not hold. While the size of the susceptible population at risk is very difficult to know in real epidemics, it is never infinite. The sequencing ratio $\rho_{SD}$ is often known to some extent, but there can be uncertainty in the number of diagnosed but not sequenced, leading to uncertainty in $\rho_{SD}$.

To investigate if the susceptible population size and biased $\rho_{SD}$ affect the estimation of the model parameters, we tested the inference method under a relative realistic scenario, i.e. setting the population sizes $N = 1000$ and introducing 10 percent bias in $\rho_{SD}$ both upwards and downwards. As before we began with one infection and stopped when there were 100 diagnosed individuals. When the simulations were stopped, the average number of infected but not yet diagnosed was 120, that is, the average prevalence was $220/1000 = 22\%$ which is a relatively higher level in reality [29].

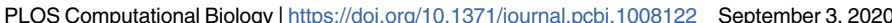

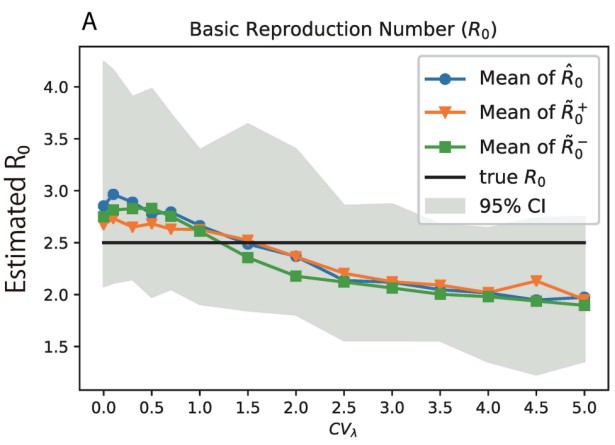
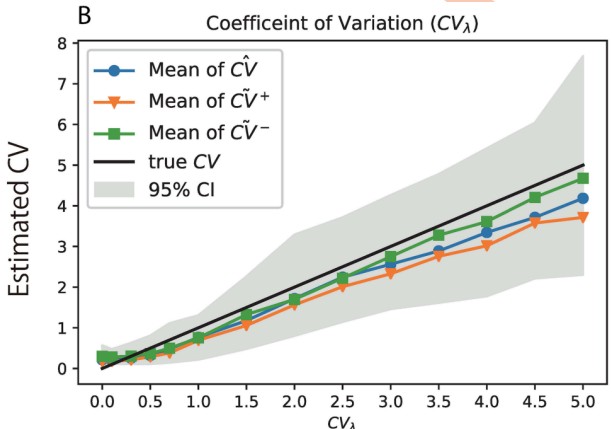

**Fig 4. Performance of parameter estimation under finite population size and biased $\rho_{SD}$.** The estimates equipped with "hat" (e.g., $\hat{y}$) are based on the infinite population size and the exact value of $\rho_{SD}$. The estimates equipped with "tilde" (e.g., $\tilde{y}^+$ and $\tilde{y}^-$) are based on finite population size and 10% bias in $\rho_{SD}$ ($\tilde{y}^+$ is on upwards bias and $\tilde{y}^-$ is with downwards bias in $\rho_{SD}$). The colored curves in (A) and (B) are the mean of 200 simulations, while the shaded areas denote the 95% confidence intervals based on these simulation replicates.

The estimation of transmission heterogeneity was robust to finite populations at risk (Fig 4) and bias in $\rho_{SD}$, with only small effects unless the true level of heterogeneity was very high ($CV_\lambda > 3$). The effects were not significant as they was covered by the variability of the estimate under infinite population size. The same conclusion applied also to estimation of the basic reproduction number $R_0$. This is encouraging, as the susceptible population size is typically unknown, and may vary over time in real epidemics.

**Inference of heterogeneity is robust to transmission heterogeneity through time.** In the above simulations, a constant transmissibility rate for each infected individual was assumed. However, it is well known that for HIV, infected individuals initially have a short high-transmission-risk phase (the acute infection phase when viral load is very high), followed by a much longer low-transmission-risk phase (the chronic phase during which viral load is relatively low), which, in the absence of treatment, is followed by another high-transmission-risk phase (progression to AIDS) [30–32].

To investigate if this transmission-risk heterogeneity through time affect the inference of heterogeneity among individuals and other model parameters, we tested the proposed inference methodology under a more realistic simulation scenario by allowing for a time-varying transmissibility rate for each infected individual. As before, each individual $i$ drew a random transmission rate $\lambda_i^0$ from the *Gamma* distribution with mean $\mu_\lambda$ and standard deviation $\sigma_\lambda$, and a duration of the infectious period being $Exp(\gamma)$ distributed. The time-varying transmission rate was modeled by starting with transmission rate $\lambda_i^0$ during acute phase and then dropping and rising again if not yet diagnosed. More specifically, we followed the four-phase model in [31] to model the change of transmission rate over time since infection that is,

$$\lambda_i(t) = \begin{cases} \lambda_i^0 & 0 - 6 \text{ months} \\ \lambda_i^0 * 1.5/8 & 6 - 15 \text{ months} \\ \lambda_i^0 * 1/8 & 16 - 36 \text{ months} \\ \lambda_i^0 * 3.6/8 & \text{after 37 months} \end{cases} \tag{10}$$

where $t$ was the time since infection for the $i$-th individual. As before each simulation was

started with one infected individual and stopped when there were 100 diagnosed individuals. The results are summarized in S4 Fig.

Relevant to the estimate of $CV_\lambda$ (i.e. heterogeneity of individual-level transmissibility) which was shown in S4B Fig, we found that adding transmission-risk heterogeneity due to disease progression leads to a small overestimation when the true level of $CV_\lambda$ was low ($CV_\lambda < 1.5$) and a slight underestimation when $CV_\lambda > 1.5$. These effects, however, were not significant as they were covered by the 95% confidence interval. In addition, as shown in S4A Fig, adding heterogeneity through time tended to underestimate $R_0$, and the bias became significant under lower level of heterogeneity ($CV_\lambda < 1.0$). This makes sense because adding heterogeneity through time reduced the average transmission rate over the whole infectious period for each individual, and hence resulted in a drop of the estimate of basic reproduction number.

**Within-host diversity causes underestimation of transmission heterogeneity.** Because HIV, and many other rapidly evolving pathogens, accumulate significant levels of within-host diversity, pathogen phylogenies may differ from the transmission history [21]. Using a coalescent-based framework that allows for within-host pathogen diversification over time [11, 21, 33], we investigated the effect of different diversification rates on the ability to detect transmission heterogeneity (please see S1 Text for details of simulation). We found that within-host diversity leads to underestimation of the level of transmission heterogeneity (Fig 5A), which becomes significant at very high heterogeneity levels ($CV_\lambda > 3$). Within-host diversity adds an additional source of phylogenetic variation, and this variation partially overshadows the heterogeneity resulting from differences in transmission contacts. Nevertheless, even with within-host diversity, the trend of the inferred heterogeneity is monotonically increasing as the true level of heterogeneity increases, leading to a positive response of the inferred level of heterogeneity. Hence, when a pathogen generates significant levels of within-host diversity, we still detect increased levels of transmission heterogeneity if it occurs, but the true level of transmission heterogeneity may become underestimated.

Increasing levels of within-host diversity also had an effect on the estimated $R_0$ (Fig 5B). While increasing levels of within-host diversification tended to increase the estimated $R_0$, only at the highest level of within-host diversity ($r = 1$) did the overestimation become significantly higher than under no diversification. As within-host diversity pushes the phylogenetic node

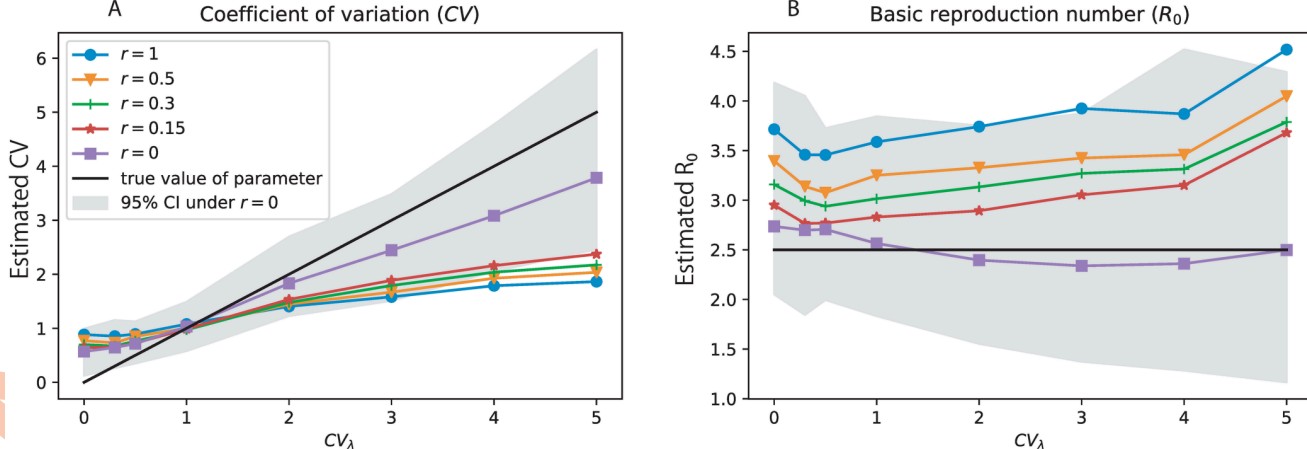

**Fig 5. Performance of parameter estimation in the presence of within-host diversity.** Here $r$ denotes the rate of within-host population size increases (per day). Four levels of within-host diversity have been calculated: $r = 1$(blue), 0.5(orange), 0.3(green), 0.15(red), and $r = 0$ (purple) corresponding to no heterogeneity. Results are the mean of 100 simulations. Also the estimates (purple) and the corresponding 95% confidence intervals (shaded area) under the condition without within-host diversity are also calculate for comparison.

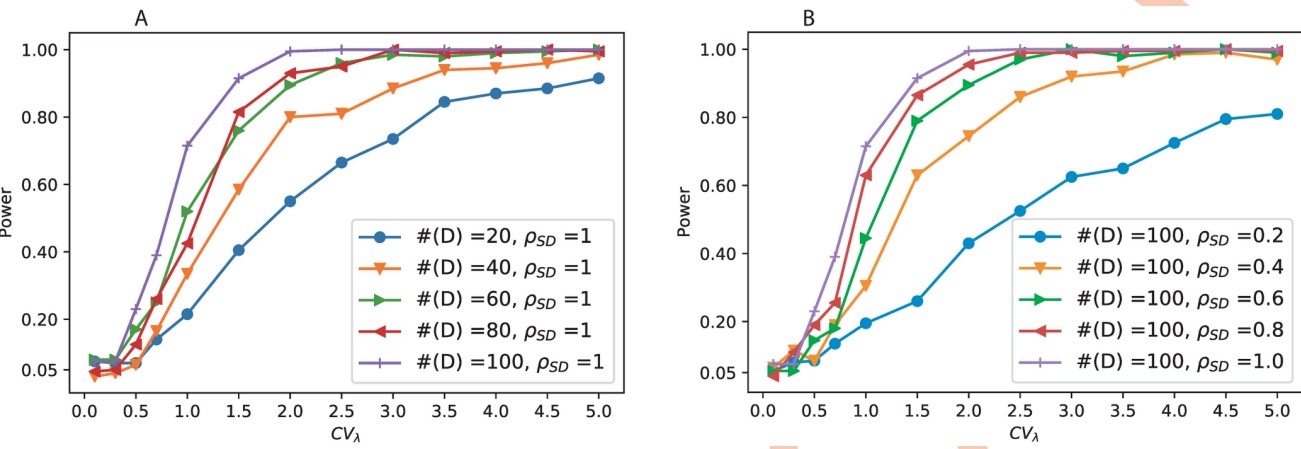

**Fig 6. Comparison of power of detecting heterogeneity under different sample sizes.** Here #($D$) denotes the number of diagnosed individuals. A. The sequencing ratio is fixed as 1, while the simulation stopped when there were different number of diagnosed individuals. B. The simulation was stopped when there were 100 diagnosed individuals (#($D$) = 100), while the sequencing ratio increases from $\rho_{SD}$ = 0.2 to 1, corresponding to a sample size going from 20 to 100 (colored curves). The power is estimated based on 200 simulations.

heights backwards in time, known as the pre-transmission interval [34], the mean infectious period ($\gamma^{-1}$) appears prolonged. However, the average time interval between neighboring transmission events, which corresponds to the mean transmissibility rate $\mu_\lambda$, is less affected by the within-host diversity. Therefore, the estimated $R_0 = \mu_\lambda/\gamma$ in the presence of within-host diversity will increase with the level of diversification.

**The power of detecting transmission heterogeneity increases with heterogeneity level and sample size.** For $\mu_\lambda$ = 1 and $\gamma$ = 1/2.5, we estimated the expected power to detect heterogeneity for various sample sizes and heterogeneity levels. We reject homogeneity if the estimated heterogeneity $\hat{CV}_\lambda$ exceeds the upper-bound of the 95% confidence interval which was estimated under the homogenous situation by setting $CV_\lambda$ = 0. This upper-bound was calculated from 500 simulations.

The power to detect transmission heterogeneity depends on the sample size and true level of transmission heterogeneity (Fig 6). We investigated two different sampling size effects; 1) the sequencing ratio at the time of stopping simulation (when 100 persons have been diagnosed), and 2) with a fixed sequencing ratio $\rho_{SD}$ at a time when a certain number of persons have been diagnosed (from 20 to 100 persons). The effects on detection power of these two sampling scenarios were similar. For a given sample size (i.e. a particular colored line in Fig 6), the power naturally increased with the level of heterogeneity. From a public health action point of view, this is again encouraging as higher heterogeneity levels suggest that intervention may be more beneficial, which we address further below. Naturally, the power also increased with larger sample size (#($D$) in panel A, and $\rho_{SD}$ in panel B).

**Computing time.** There is a growing demand of analyzing large sequence databases where transmission heterogeneity may exists and be desirable to detect [35]. Hence, we evaluated the computing time required to process trees with the number of tips (individuals) varied from 100 to 4,000. These results are summarized in supplementary S8 Fig. Our result indicated that the average computing time (over heterogeneity level $CV_\lambda$ varying from 1 to 5) scales linearly with the size of the tree. For instance, it required about 54.8 minutes (averaging over heterogeneity levels from 1 to 5) to process a tree with 4,000 tips. We also observed that the computing time for a tree with low level of heterogeneity (i.e. $CV_\lambda$ = 1) is much longer than that of analyzing a tree with high level of heterogeneity (i.e. $CV_\lambda \geq 3$). And the difference

between these two computing times increases with the size of the tree. This is due to the fact that the proposed algorithm needs to reconstruct the transmission history while estimating the model parameters. This reconstruction is easier to proceed under the situation with high level of heterogeneity where few super-spreaders contribute to most of the transmission events. Hence, we expect that more computing time is needed to process a tree with heterogeneity smaller than 1 ($CV_\lambda < 1$).

## Analysis of real HIV outbreak data

We investigate HIV-1 DNA sequences from three local epidemics in Sweden ranging from fast explosive outbreak to more slow spread over longer time periods (Fig 7). The three corresponding genealogies were selected from a recent publication on HIV-1 spread in Sweden and neighboring Scandinavian countries [36], but the datasets used here contain only Swedish sequences. The first genealogy, denoted IDU_AE, represents a rapid outbreak of CRF01_AE (circulating recombinant form number 1 of subtype A and E viruses) infections among intravenous drug users (IDUs) in Stockholm in 2006 and 2007 [37]. The genealogy contains 83 taxa which were sampled from 2003 to 2010. The second genealogy, denoted IDU_B, represents slower local spread of subtype B infections among IDUs in Stockholm with sampling dates ranging from 2003 to 2010 [38]. This genealogy contains 51 taxa which were sampled from 2002 to 2009. The last genealogy, denoted MSM_B, represents longstanding local spread of subtype B virus among men who have sex with men (MSM) with sampling dates ranging from 1994 to 2010. This genealogy contains 18 taxa which were sampled from 1994 to 2010. All genealogies were monophyletic relative to other sequences in the Swedish-Scandinavian study and BLAST'ed international sequences [36].

For each data set, we analyzed a random sample of 10000 genealogies from the posterior distribution (generated using BEAST [39]). Fig 7 summarizes the results of the parameter estimates. Overall, the IDU_B data showed the lowest diagnosis rate ($\hat\gamma$), meaning the individuals infected in this outbreak had a longer transmission period before being diagnosed (and prevented to spread further). Interestingly, the IDU_B outbreak also showed the highest $R_0$, suggesting that the low value of $\hat\gamma$ may have driven up $R_0 = \mu_\lambda/\gamma$. $R_0$ did not seem directly linked to transmission heterogeneity, however. Similarly, Sackin's index, which measures tree balance (high values indicate unbalanced trees where one side of a split carries more taxa than the other side), did not find any significant differences between the data that contained an outbreak (IDU_AE) and the data from more even spread (IDU_B). Instead, as one might have expected, $\hat{CV}_\lambda$ correlates with the speed of the different outbreaks, i.e. the highest heterogeneity was inferred in the data that contained the fast outbreak, which occurred in the IDU_AE epidemic, and the lowest heterogeneity was inferred in the slowest spreading epidemic (MSM_B). These observations agree with previous analyses of these local epidemics, where it was reported that there was a rapid outbreak, during a limited time period, among IDU infected with CRF01_AE in Sweden following a similar outbreak in Finland [38]. This suggests that it took some time for HIV-1 CRF01_AE to reach one or several individuals with high contact numbers after it had entered the Swedish IDU community. Once the individuals with high contact rates were reached, a rapid outbreak followed, infecting many susceptibles. Finally, as shown in [38], the outbreak subsequently slowed down to pre-outbreak levels, presumably because the recipients of the high-contact persons had been exhausted. Lower levels of transmission heterogeneity was observed in the slower spread that took place in the IDU_B community [37], suggesting that the contact patterns in this community were more even among members.

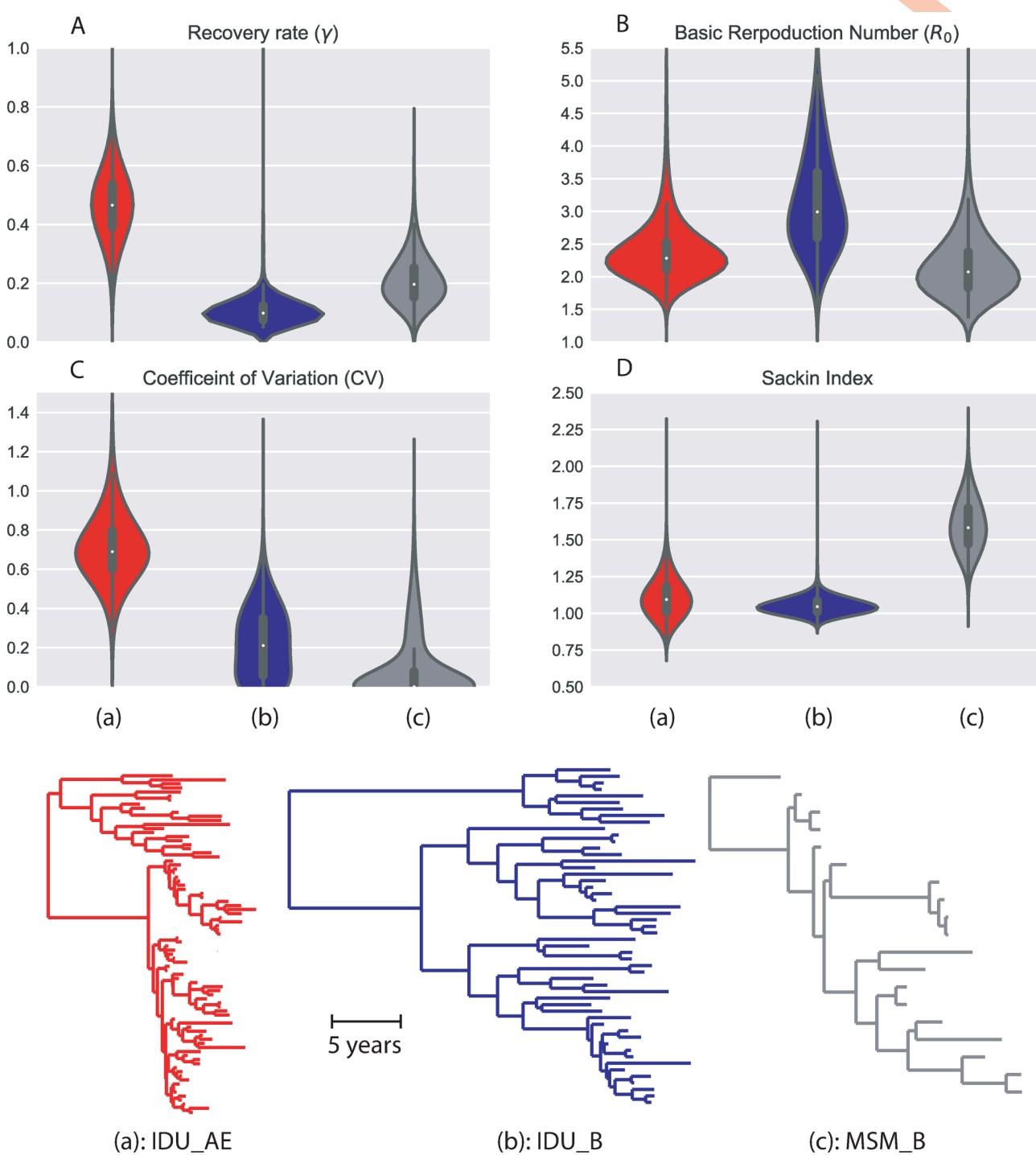

**Fig 7. Estimation of heterogeneity and other epidemiological parameters for three HIV outbreaks in Sweden: IDU_AE (red), IDU_B(blue) and MSM_B (grey).** On the top four panels, the distributions of the estimated parameters from a sample of 10000 posterior genealogies of each dataset are presented: the rate of diagnosis (A), the basic reproduction number (B), the Coefficient of Variation (C) and the *Sackin* Index normalized with the 'PDA' method (D). On the bottom, the time-scaled genealogies with the highest posterior probability are presented.

## Simulation of targeted contact tracing: Continuous monitoring and cross-sectional sampling

Next, we evaluated phylogeny-guided contact tracing policies based on simulation. We again set the mean transmissibility rate at $\mu_\lambda = 1$ and the rate of diagnosis at $\gamma = 1/2.5$, which resulted in the basic reproduction number as $R_0 = \mu_\lambda/\gamma = 2.5$. Here, one unit of time corresponded to one year as before, and we set the time period $t_c$ in which we calculate the *NCE*-value to $t_c = 1.25$ (corresponding to half of the average infectious period). In addition, the heterogeneity varied from $CV_\lambda = 0$ to 5.

We define 'continuous monitoring' as the typical public health situation where samples (sequences of diagnosed cases) are continuously collected throughout an epidemic and where for each collected sample it is decided to perform (additional) contact tracing or not. The second situation is denoted 'cross-sectional sampling' which is the situation where all samples are collected (or re-analyzed) at some common time $t$, for instance when the epidemiological status is investigated in an otherwise unmonitored population or when additional preventive measures are decided upon or in retrospective contact tracing.

**Using a pathogen phylogeny improves disease prevention under continuous monitoring.** When no transmission heterogeneity is present ($CV_\lambda = 0$), targeting individuals with higher than average *NCE* for contact tracing has no effect on the resulting epidemic size (Fig 8A), i.e. randomly selecting the same number of persons for contact tracing has the same effect as a phylogenetically informed strategy (the dashed and solid lines are on top of each other in this case). The reason why preventing (i.e. contact tracing) persons with $NCE \geqslant 1$ has higher effect than $NCE \geqslant 2$ is simply because $NCE \geqslant 1$ involves a larger number of persons that are contact traced, shown in the sidebar 'Fraction of contact traced'. The 'no prevention' line shows the epidemic growth when no prevention (i.e. no additional contact tracing) was administered and is thus the maximum size of the simulated epidemic at any time point. Reduction from this level indicates the prevention effect.

When transmission heterogeneity was present (i.e. $CV_\lambda > 0$), the relative epidemic size under a phylogeny-guided prevention (based on *NCE*) was always smaller than that of random prevention at the same proportion of persons (Fig 8B–8D). At very high level of heterogeneity $CV_\lambda = 3$, phylogeny-guided prevention typically reduced the epidemic by an additional 10% over random prevention at 2.5 years out, and approximately an additional 10% for each *NCE* step. Moreover, the advantage of a $NCE \geqslant m$ strategy over the corresponding random prevention strategy increased with the level of heterogeneity. For instance, if extremely high heterogeneity occurred ($CV_\lambda = 5$), by preventing spread at $NCE \geqslant 4$ one would only need to contact trace 20% of the population to get about 75% reduction in total number of infections 2.5 years later. This means that a phylogenetically informed prevention strategy (using the *NCE* concept) becomes more valuable the more heterogeneous transmission rates are.

From a public health point of view, where resources always are limited, using the pathogen phylogeny to allocate resources becomes more efficient the more heterogeneity that exists in the epidemic, and the *NCE* guided prevention can pinpoint where contact tracing and prevention would make the largest impact in reducing future infections. This becomes clear when comparing the fraction of contact traced and the relative effect of *NCE*-guided prevention in the different $CV_\lambda$ and *NCE* cases, i.e. the fraction of contact traced in the $NCE \geqslant m$ strategies decreased as the threshold value $m$ increased while the relative effect over random prevention in the $NCE \geqslant m$ strategies increased as the threshold value $m$ increased (see the supplementary S5 Fig). Therefore, under low levels of heterogeneity (estimated using our general heterogeneity detection method), one may choose a small threshold $m$ to gain more effect on reducing

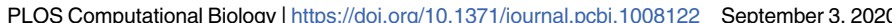

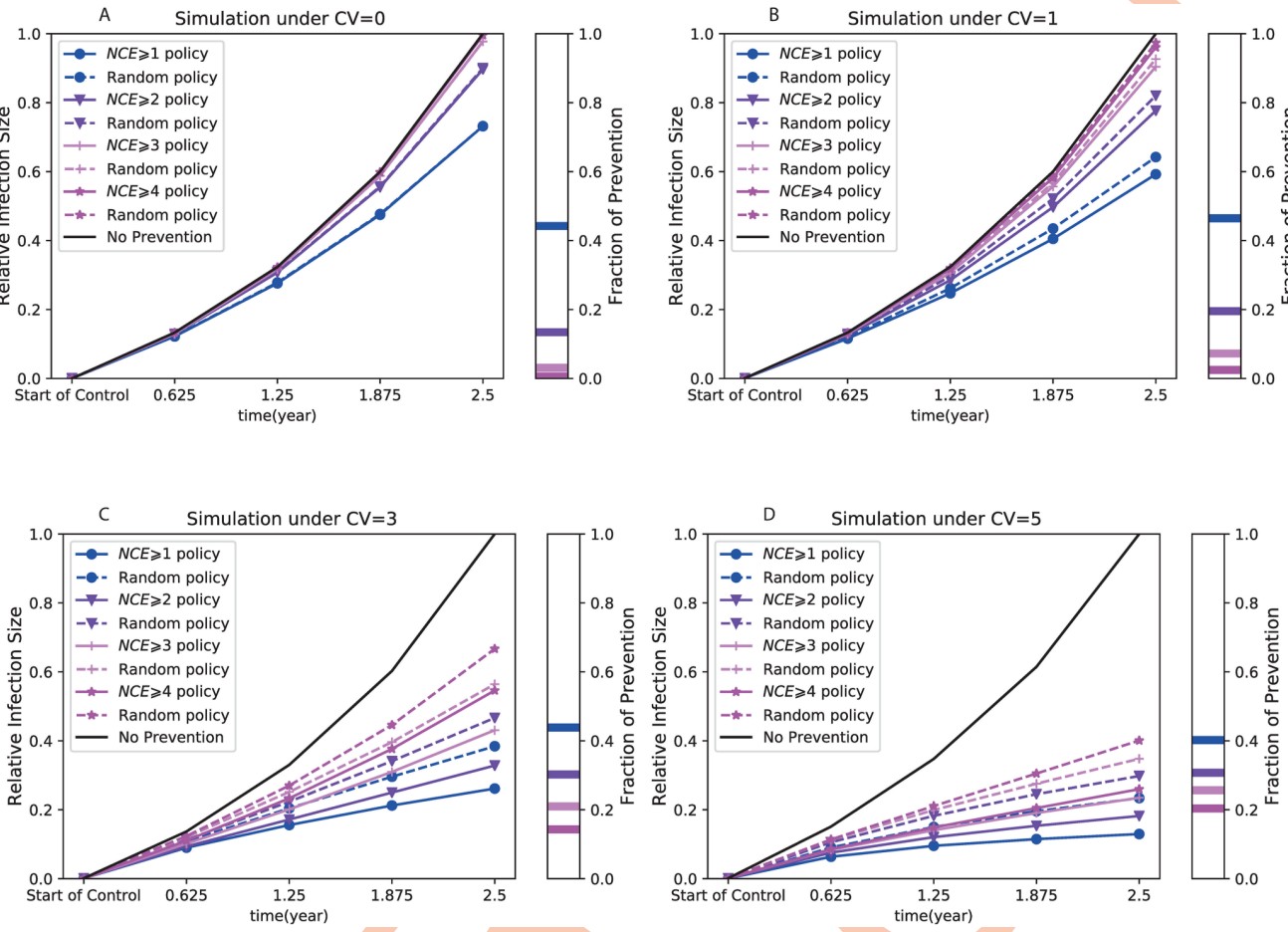

**Fig 8. Comparison of *NCE*-based contact tracing and random contact tracing under the situation of continuous monitoring.** The relative infection sizes (defined in S1 Text) of the *NCE* ≥ *m* contact tracing (solid lines) and the corresponding random contact tracing (dashed lines) are compared for four threshold values: *m* = 1 (lines with ●), *m* = 2 (lines with *triangledown*), *m* = 3 (lines with +), and *m* = 4 (lines with *). The subplots on the right side of all panels show the fraction of contact traced for all these prevention policies. Four panels show the comparison under four levels of heterogeneity respectively, i.e. $CV_\lambda = 0$ (A), $CV_\lambda = 1$ (B), $CV_\lambda = 3$ (C), and $CV_\lambda = 5$ (D). These results are the mean of 300 simulations.

the epidemic size, while under high levels of heterogeneity, one may instead choose a high threshold *m*, involving less resources, to invoke efficient control.

For pathogens that accumulate within-host diversity, such as HIV-1, we also evaluated the phylogeny-guided prevention strategy when significant within-host diversity accumulates. The effect of a *NCE*-strategy was, in fact, higher in the presence of within-host diversity (*r* = 1) than when no within-host diversity accumulated (*r* = 0) (Fig 9). This happens because large *NCE* values are associated with short branches, and the effect of within host diversity is to make short branches only slightly longer whereas longer branches are increased more due to within host diversity. As a consequence, within-host diversity has the effect of accentuating the difference between short and long branches thus leading to a better distinction between subtrees with a relatively large number of coalescent events (large *NCE*) compared to subtrees with lower number of coalescent events.

Within-host diversity also has the effect of reducing the fraction of contact traced for given threshold *m* in *NCE* (S6 Fig). Because within-host diversity pushes the phylogenetic node heights backwards in time, known as the pre-transmission interval [34], there will be fewer nodes at any given time-distance into the tree from the tips. Within-host diversity therefore

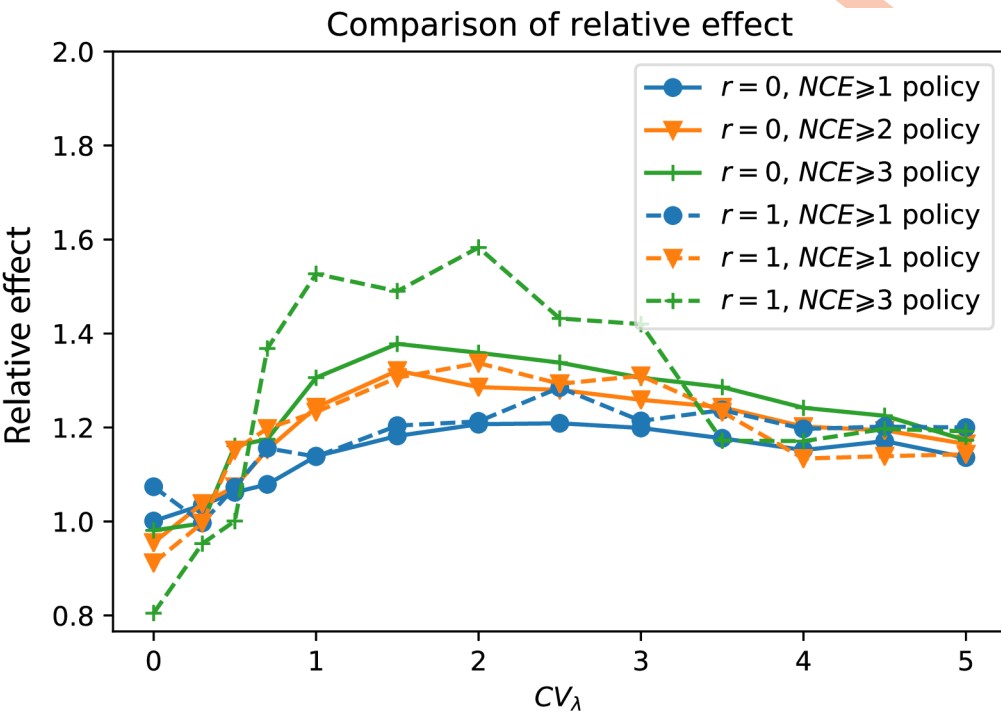

**Fig 9. Comparison of the performance of *NCE*-based policies under the situations of with/without within-host evolutionary dynamics.** Here $r$ is the diversification rate of within-host virus population. The solid lines ($r = 0$ lineage/day) and the dashed lines ($r = 1$ lineage/day) are the results with/without within-host diversity respectively. The performance of the $NCE \geqslant m$ policy is measured by the relative effect over random prevention (please see text for definition). Three levels of threshold $m$ are calculated: $m = 1$ (blue), $m = 2$ (orange), and $m = 2$ (green).

reduces the *NCE*-count on all lineages, leading to a reduction in the fraction of contact traced. Hence, one may adjust the depth $t_c$ at which *NCE* is calculated depending on how much diversity the pathogen under study typically accumulates.

Finally, we evaluated the phylogeny-guided prevention under two different models of contact tracing: single-step tracing (where only direct recent contacts of the diagnosed are tested and prevented from further spread) or iterative tracing (where new cases from the single-step tracing are also contact traced, and so on) [24, 40]. The relative effect of phylogeny-guided iterative contact tracing was more effective than single-step contact tracing at higher levels of the *NCE*-threshold ($m = 2$ and 3) in Fig 10. Contrary, at $NCE \geqslant 1$, single-step contact tracing did better at lower levels of transmission heterogeneity. Hence, iterative contact tracing is more advantageous when medium to high levels of transmission heterogeneity exist, where higher *NCE*-thresholds can pick up transmission chains where a super-spreader exists. At very high heterogeneity levels ($CV_\lambda \geqslant 3$), the iterative contact tracing advantage gradually diminishes because the probability to find the super-spreaders increases after just a single-step contact tracing.

As somewhat surprizing observation was that the fraction of contact traced under iterative contact tracing was actually found to besmaller than under the single-step contact tracing (S7 Fig). For each selected individual to contact trace there will of course be more (or at least not fewer) to contact trace under iterative contact tracing. However, iterative contact tracing has an even bigger negative effect on the spreading, such that the overall fraction (or number) being contact traced in the iterative setting is smaller than when applying single-step contact tracing.

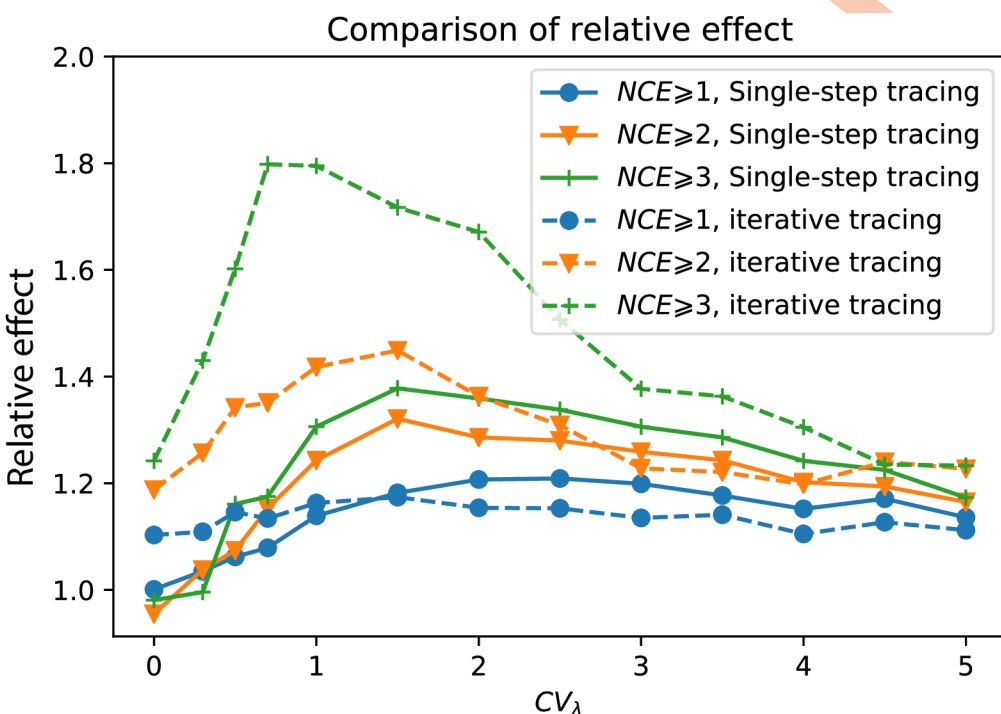

**Fig 10. Comparison of the performance of *NCE*-based policies under different modes of prevention.** The solid lines/dashed lines are the results under the mode of single-step contact tracing/iterative contact tracing respectively. The performance of the *NCE* ⩾ *m* policy is measured by the relative effect over random prevention (please see text for definition). Three levels of threshold *m* are calculated: *m* = 1 (blue), *m* = 2 (orange), and *m* = 2 (green).

**Time-to-diagnosis is informative for contact tracing under cross-sectional sampling.** When cross-sectional samples have been collected, phylogeny-guided prevention has no advantage over a random prevention strategy (Fig 11). Thus, to take advantage of the epidemic information captured by the pathogen phylogeny, continuous monitoring provides better public health prevention capacity. If only a cross-sectional sample is available, however, we find that targeting the most recently diagnosed individuals (*MRD*) for additional contact tracing performs better in preventing future spread than a random selection of the same number of individuals (Fig 11) or by selecting individuals to contact trace based on the *NCE*-value. At prevention fractions *p* = 0.2 and *p* = 0.5 (20% and 50% being contact traced, respectively), the relative effect of the *MRD*-strategy was constantly larger than random selection (of which the relative effect was set to 1) as well as that of the *NCE*-based strategy.

## Discussion

In this study we have shown that when transmission heterogeneity exists, a phylogeny-guided prevention strategy is more efficient than randomly selecting cases for contact tracing in order to reduce the number of future infections. We first developed a general method to identify the level of transmission heterogeneity in a population, i.e. how much variation in the rate of transmission contacts that exist in a human population where a pathogen spreads. We then developed a phylogenetic measure, the *NCE*—the Number of Coalescent Events over a fixed time-interval ending at time of diagnosis—to identify which infected individuals to target for (additional) contact tracing. It was shown that, under the typical situation where pathogen samples are continuously collected (continuous monitoring), a phylogeny-guided prevention

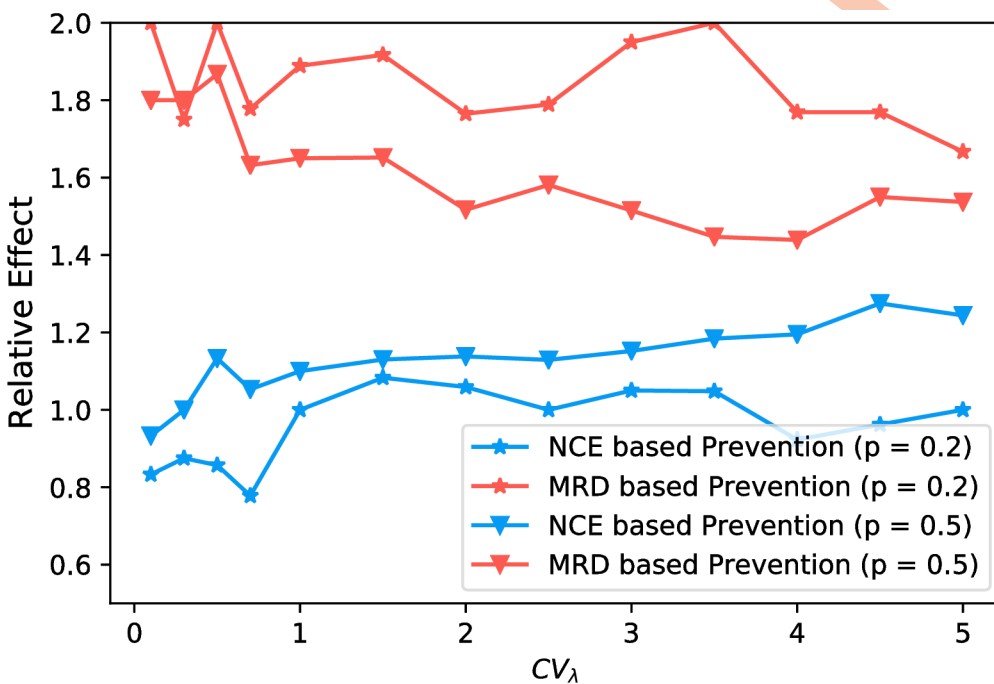

**Fig 11. Relative effect of *NCE*-based strategy (blue lines) and MRD strategy (red lines) over random prevention under the situation of Cross-sectional prevention.** The fraction of contact traced is fixed as $p = 0.2$ (lines with $\triangledown$) and $p = 0.5$ (lines with $\star$).

approach was clearly superior in effectively allocating public health resources to reduce future disease spread.

We show that our heterogeneity detection method and phylogeny-guided approach work under a variety of realistic conditions. The estimates of heterogeneity and basic reproduction number are robust to assumptions about the susceptible population size and exact sampling fraction (the fraction of diagnosed individuals that are sequenced), both of which may be difficult to know exactly in a real epidemic. Also when changing the recovery rate of the infected individual, allowing autocorrelation in transmission rates, or adding heterogeneity through time to individual-level transmissibility, the estimate of heterogeneity is still less affected. Of particular importance is that our method also works when a pathogen accumulates genetic variation within each host. Such within-host diversity is significant in e.g. HIV infections. If not taken into account, tree measurements may significantly mislead epidemiological estimates by implicitly assuming identity between transmission history and pathogen phylogeny. It is well established that the between-host pathogen phylogeny will display nodes corresponding to the transmitted lineages biased backwards in time, as described by the pre-transmission interval [34], as well as potential lineage disordering relative to the transmission history [21]. While previously often ignored, the within-host diversity has recently been shown to have significant impact on inferences about pathogen epidemics, sometimes rendering results completely as compared to when within-host diversity was ignored [11, 21, 41, 42]. Here, we show that taking within-host diversity into account may, in fact, improve the prevention effect (Fig 9).

When applying our method for detecting possible spread heterogeneity to three separate sub-epidemics from the larger Swedish HIV-1 epidemic it was found that the highest heterogeneity was inferred from the data that included the episodic IDU_AE outbreak. Interestingly,

this outbreak did not show significant differences from the other sub-epidemics when analyzing Sackin's index or $R_0$ (which is also affected by the estimated diagnosis rate $\hat{\gamma}$). This suggests that our methodology is useful for identifying heterogeneous spread when typical tree balance or well-known epidemiological parameters fail.

In some countries and settings where HIV transmission is criminalized there may be ethical limitations to the use of phylogenetics and other ways to identify clusters with active HIV transmission. We acknowledge this complication, but show that our approach could limit HIV transmissions in settings where such limitations do not exist.

Our model treats heterogeneity as a single feature, while in real life there are several factors that may contribute to heterogeneous spread rates. It is possible that modeling one or several of these factors separately may be valuable. One such factor relates to social network structure, modeling heterogeneities in terms of contacts. Secondly, the current model assumes that the transmissibility of different individuals are independent. A (more complicated) alternative could be to assume that transmissibility of infectors and their cases are positively correlated, which would imply assortativity in sexual activity between sex-partners. Finally, individuals may also differ in terms of testing (i.e. diagnosis) rates; in our model all individuals were assumed to have equal rates of testing. For example, among MSM, individuals with risky behavior (high contact rates) may also test themselves more often. This simplifying assumption may lead to a systematic loss of sensitivity as sampling may not be complete and be biased towards some group of infected persons. Again, the heterogeneity in our model may be interpreted as a combination of all these and other aspects.

In conclusion, with the increase of pathogen genetic data in public health databases, phylogeny-guided prevention will benefit public health efforts. As relevant epidemic information about differences in number of contacts is effectively "recorded" in a phylogeny by the evolutionary process of the pathogen, it only makes sense to use this information to the public health benefit. Hence, phylogeny-guided prevention results in more efficient epidemic control, which should reduce disease burden and costs to society.

## Supporting information

**S1 Fig. Average branch length of the simulated tree under different conditions.** Average branch length of the simulated tree under varied basic reproduction number (x-axis) and varied levels of heterogeneity (y-axis). Under each condition, the simulation started from one infection and stopped when there were 100 recovered individuals. The average branch length was calculated as the total branch length of the reconstructed tree over the number of tips. The results were the average over 1000 simulations.
(PDF)

**S2 Fig. Inference of heterogeneity under various lengths of mean infectious period $\gamma^{-1}$.** In each panel, the black line denotes the true value that was used to generate the simulated data. The colored curves are the means of the estimates under different levels of $\gamma^{-1}$. The shaded area denotes 95% confidence interval estimated when $\gamma^{-1} = 2.5$. These results are obtained from 100 simulation replicates where the average transmissibility rate $\mu_\lambda$ was fixed as 1, the sequencing ratio $\rho = 0.9$, and the simulation stopped when there were 100 diagnosed individuals. The panel A is coefficient of variation ($CV_\lambda$) and the panel B is standard deviation of infectivity rate ($\sigma_\lambda$).
(PDF)

**S3 Fig. Performance of parameter estimation under various lengths of mean infectious period $\gamma^{-1}$.** In each panel, the colored curves are the means of relative biases (the bias over the

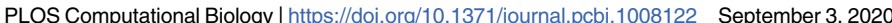

true value) under different levels of $\gamma^{-1}$. The shaded area denotes the relative bias of the 95% confidence interval estimated when $\gamma^{-1} = 2.5$. These results are obtained from 100 simulation replicates where the average transmissibility rate $\mu_\lambda$ was fixed as 1, the sequencing ratio $\rho = 0.9$, and the simulation stopped when there were 100 diagnosed individuals. A. basic reproduction number ($R_0$), B. average infectivity rate ($\mu_\lambda$), and C. recovery rate ($\gamma$).
(PDF)

**S4 Fig. Comparison of parameter estimation under constant/time-varying transmissibility.** The lines with circles "∘" are the estimate under the situation of constant transmissibility, and the shaded area denote the 95% confidence interval from 100 simulations. The lines with triangles "▽" are the estimates under the situation with time-varying transmissibility (TVT). The comparison is performed under the situation of $R_0 = 2.5$ and the heterogeneity ($CV_\lambda$) varies from 0 to 5. Sample size n = 100 and simulation runs = 100.
(PDF)

**S5 Fig. Relative effect over random prevention of $NCE_m$ strategy under the continuous monitoring scenario.** Four levels of threshold $c$ have been calculated: $m = 1$ (blue), $m = 2$ (orange), $m = 3$ (green), and $m = 4$ (red). Results in (a) and (b) are the mean of 300 simulations.
(PDF)

**S6 Fig. Comparison of fraction of contact traced of $NCE_m$ strategy under the situation of with/without within-host diversity (solid/dash lines respectively).** Three levels of threshold $m$ have been calculated: $m = 1$ (blue), $m = 2$ (orange), and $m = 3$ (green). Results are the mean of 300 simulations.
(PDF)

**S7 Fig. Comparison of fraction of contact traced of $NCE_m$ strategy under the situation of single-step/iterative contact tracing (solid/dash lines respectively).** Three levels of threshold $m$ have been calculated: $m = 1$ (blue), $m = 2$ (orange), and $m = 3$ (green). Results are the mean of 300 simulations.
(PDF)

**S8 Fig. Computing times required for inferring heterogeneity from trees with varying numbers of tips.** Computing times (y-axis) required for processing trees with varying numbers of tips (each tip represents a diagnosed individual). For each number of tips (x-axis), we simulated trees with different levels of heterogeneity (i.e., $CV_\lambda = 1, 2, \cdots, 5$). Each point represents the average computing time based on 10 simulations with given heterogeneity level and the size of tree. The number of tips was varied along the following sequence: 100, 200, 400, 1000, 2000, 4000. The dashed line represents the average computing time over all levels of heterogeneity. All runs were executed on an Intel Core i7 processor (Mac mini 2018).
(PDF)

**S9 Fig. Performance of new method and the Markov-modulated Poisson process (MMPP) based genetic clustering method on simulated data.** Virus genealogies were simulated under two scenarios: without autocorrelation in transmission rates (A and B) and with autocorrelation (C and D). In the former cases, each infected individual independently draws its transmission rate from a binomial distribution, i.e., choosing the slow rate $\lambda_1$ with probability $1 - \pi_c = 0.9$ and choosing the fast rate $\lambda_2$ with probability $\pi_c = 0.1$. In (A), $\lambda_1 = 2$ and $\lambda_2 = 6$, corresponding to mean rate of $\mu_\lambda = 2.4$ and a low level of heterogeneity ($CV_\lambda = 0.5$). In (B)$\lambda_1 = 1$ and $\lambda_2 = 15$, corresponding to mean rate of $\mu_\lambda = 2.4$ and a high level of heterogeneity ($CV_\lambda = 1.75$). In the cases with autocorrelation in transmission rates, each newly infected individual

                                    

switch its transmission rate with probability $\pi_s$ and remains its infectee's rate with probability $1 - \pi_s$. In both (C) and (D), $\lambda_1 = 0.9$ and $\lambda_2 = 8.1$ with different switching probability, i.e., $\pi_s = 0.2$ in (C), corresponding to $\mu_\lambda = 5.22$ and low level heterogeneity $CV_\lambda = 0.79$, and $\pi_s = 0.8$ in (D) corresponding to $\mu_\lambda = 2.64$ and low level heterogeneity $CV_\lambda = 1.27$. And the diagnosis rate is fixed as $\gamma = 1$. The x- and y-axes correspond to the estimated $\mu_\lambda$ and the estimated $CV_\lambda$ respectively. Each point represents the outcome when analyzing one of 100 replicates. (PDF)

**S1 Text. Supporting information on models, simulation methods and the phylogenetic analysis of HIV data in Sweden.** Includes details on simulating the virus genealogy, how the correction for incomplete transmission chain were derived, the preprocess of the real epidemiological data from Sweden, how the evaluation of control measures was performed based on simulation, and the pseudo-code of the proposed algorithm. (PDF)

## Author Contributions

**Conceptualization:** Jan Albert, Tom Britton.

**Data curation:** Yunjun Zhang, Jan Albert.

**Funding acquisition:** Tom Britton.

**Methodology:** Yunjun Zhang, Thomas Leitner, Tom Britton.

**Project administration:** Tom Britton.

**Resources:** Jan Albert.

**Software:** Yunjun Zhang.

**Supervision:** Tom Britton.

**Writing – original draft:** Yunjun Zhang, Thomas Leitner, Tom Britton.

**Writing – review & editing:** Yunjun Zhang, Thomas Leitner, Tom Britton.

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
