## [Decision Letter · Decision Letter 0]

6 Nov 2019

Dear Dr zhang,

Thank you very much for submitting your manuscript 'Inferring transmission heterogeneity using virus genealogies: estimation and targeted prevention' for review by PLOS Computational Biology. Your manuscript has been fully evaluated by the PLOS Computational Biology editorial team and in this case also by independent peer reviewers. The reviewers appreciated the attention to an important problem, but raised some substantial concerns about the manuscript as it currently stands. While your manuscript cannot be accepted in its present form, we are willing to consider a revised version in which the issues raised by the reviewers have been adequately addressed. We cannot, of course, promise publication at that time.

Sincerely,

Roger Dimitri Kouyos

Associate Editor

PLOS Computational Biology

Virginia Pitzer

Deputy Editor

PLOS Computational Biology

[LINK]

Reviewer's Responses to Questions

**Comments to the Authors:**

Reviewer #1: This manuscript describes a new method to detect variation in transmission rates from the shape of the phylogenetic tree reconstructed from pathogen sequences. This quantity has public health relevance because it may enable one to optimize the allocation of prevention resources by focusing on those individuals with the highest probability of onward transmission, based on past dynamics.

More specifically, the "heterogeneous birth-death" model presented by the authors is an extension of the standard birth-death model to accommodate variation in transmission rates among individuals through a random-effects approach, where these rates are drawn from a gamma distribution to be parameterized from the data. This random-effects approach is distinct from that of otherwise similar models, such as Kuhnert’s two-type birth-death model (PLOS Pathog 14(2): e1006895), Barido-Sottani’s multi-state birth-death model (J R Soc Interface 15(146)) or McCloskey’s pure-birth model (PLOS Comput Biol 13(11): e1005868), where lineages “evolve” between latent discrete branching rate states. Notably, none of these publications are cited by the authors. (See also a preprint from Volz et al.; bioRxiv, https://doi.org/10.1101/704528). Since the authors are presenting a new method, they should also evaluate their method alongside at least one of these other approaches that are generally designed to accomplish the same objective.

Furthermore, the rates in the heterogeneous model are distributed among individuals in an uncorrelated fashion with respect to the transmission history, whereas the rates are autocorrelated in the two-type and multi-state models. This difference highlights the focus of this manuscript on the individual host, rather than groups in a risk-structured population. The assumption of uncorrelated transmission rates (absence of homophily by risk) needs to be justified (e.g., is it appropriate for HIV-1), as it is a distinctive feature of this model. The authors should also evaluate the impact of variation in diagnosis rates among individuals (rate parameter $\\gamma$) on their power to detect heterogeneity in transmission rates (more below).

An important starting assumption of the model is that transmission unfolds in an infinite population of susceptible hosts, which supports the approximation that “transmission events occur independently” (line 164), for instance. The authors evaluate their model when this assumption is relaxed - however, this was carried out in the initial exponential growth phase of the epidemic (starting from a single infected individual to an average of 120 infections in a total population of 1000 hosts). This minimizes the effect of limited susceptibles, i.e., the “collision” of infected lineages, and it would be helpful to see the robustness of the model when the epidemic is at a more intermediate stage.

Another, more problematic assumption of this model is that the expected time to diagnosis is the same across all individuals. Incomplete and inevitably biased sampling (diagnosis) is widely recognized as a severe confounding problem in measuring variation in transmission rates. For example, if individuals with higher transmission rates are also more likely to be lost to follow-up for sampling (sequencing after diagnosis), then the model (and most other models as well) will suffer from a systematic loss of sensitivity. This limitation should be addressed in the Discussion section.

While the authors evaluated the assumption of incomplete sampling by allowing for transmission to unsampled individuals, they also assume that there is no onward transmission from unsampled to sampled individuals - viz., from the Supplementary Text S1: “Furthermore, we assume that the unsampled individuals have small level of transmissibility rates and that they caused no infection before being diagnosed.” This is an unrealistic assumption and especially problematic in the case of targeting specific individuals for public health intervention – to say nothing of the potential exposure of individuals to stigma and criminalization – and it should be described more transparently in the manuscript (for instance, around lines 221-223) and rationale given.

The GitHub repository should incorporate the documentation in the Word document (which is a binary file) into the README in the conventional Markdown format. In addition, I found that the Python scripts do not adhere to PEP8 (e.g., lack of whitespace separating arguments) and there is a general lack of inline or block comments that would make it easier to review the code. Variables such as “mu_RecEst” need to be defined within the source code. There are multiple uses of undocumented magic numbers (e.g., 9999) where it is more conventional to use the Python Nonetype or math.inf. The module documentation needs to provide installation instructions, including module dependencies such as matplotlib and scipy. The tutorial is incomplete and should provide instructions for loading the module prior to calling functions such as “singleSimu”.

In my small experiments with these scripts, I found that mean $\\lambda$ was substantially underestimated when setting $\\mu$ to 3.0 - the average estimate of 5 trials was 1.57 ($\\gamma$ set to 0.3). This may be masked by reporting accuracy for CV and not the mean, particularly when the variation parameter is underestimated the same way. Please provide results for these parameters individually. There also seems to be some interaction with $\\gamma$; if I set this parameter to 0.7 then the mean estimate of $\\mu$ increased to 1.99. This effect on parameter identifiability should be explored.

I could not find details on run times of the analysis or estimates of time complexity. It took roughly 10 seconds to simulate and analyze a single tree with sample size 100 (default setting) on a rather outdated iMac (late 2013), and about 1 minute for a sample of 200, so I suspect that the complexity will be quadratic or more with sample size. An assessment of computing time should be incorporated into the revision.

As developers and users of phylogenetic methods in the context of HIV-1, we are obligated to discuss the ethical implications of work that can be potentially used for source attribution (i.e., “who infected whom”). Specifically, this study proposes to identify individuals who are associated with higher numbers of coalescent events, which can be loosely interpreted as transmission events, as targets for public health intervention. In light of ongoing criminalization of HIV in many countries worldwide, and where punishments are often disproportionately severe, it is inevitable that methods such as the one presented by the authors here will also be used in the prosecution of individuals for transmission without disclosure. The authors should address this problem in the discussion section of their revised manuscript.

Overall, I found this an interesting manuscript. It is written well and I found the model description mostly straight-forward. As discussed above, there are a number of potentially (or almost certainly) problematic model assumptions that need to be explored further, and the authors should address other work in this area and perform some comparison of methods.

- line 64: Is there any empirical motivation for using the gamma distribution to model variation in transmission rates among individuals, other than this being a relatively flexible distribution for continuous non-negative values?

- Minor point (line 94): I prefer to reserve the term “genealogy” for describing the ancestral relationships among individuals in a population, and “phylogeny” for the ancestral relationships among populations. Thus the variable G represents a phylogeny relating infections sampled from different hosts. However, I also recognize that others refer to this tree as a “genealogy” or use these terms interchangeably.

- line 102: “… or in relative terms as the coefficient of variation CV_\\lambda = \\frac{\\sigma_\\lambda}{\\mu_\\lambda}.” For readers who are less familiar with the CV, it would be helpful to note that this is a dimensionless quantity where standard deviation, as a measure of dispersion, is normalized by the mean that facilitates comparisons across data sets.

- lines 165-174: This section would greatly benefit from a diagram explaining the relationships among d_i, e_i, l_b, etc., as there is a substantial amount of notation being introduced here.

- line 369: Please provide methodological details on the BEAST analysis, i.e., model assumptions, prior distributions, assessment of convergence.

- Text S4: “During this simulation, when there is a newly diagnosed individual, the corresponding NCE value is calculated and the decision of contact tracing has been made based on the NCE value.” Is there a delay in this simulation for the time between diagnosis and sampling (sequencing), which would be required to compute the NCE value associated with that individual?

- Figure 1: The notation in the figure does not correspond to the figure legend. For example, “the i-th individual was infected at t_i” – neither symbol appears in any of the figures. It seems that the authors switch between Roman numeral notation for branches and Arabic numerals for the time axis.

- Figure 2. “The internal branches with the same label represent the infectious period of a particular individual.” But what if the start of the infectious period does not coincide with the branching point in the virus genealogy? This statement seems to assume that there is no within-host variation and that individuals are infectious immediately following transmission.

- Figure 4. The data series for D=100 and \\rho_{SD} = 1.0 should be identical between plots A and B, but they do not seem to be.

Reviewer #2: With this manuscript the authors propose a method to estimate the amount of transmission heterogeneity based on virus sequence data in a sampled epidemic and suggest useful applications for the method. While the method is very interesting and the implementation of heterogeneity is simple but novel, the range of simulated situations is rather limited, e.g. it was only tested for a single R0 value (2.5). Hence, I suggest to ensure the method is indeed applicable to a wide range of situations.

Further comments and more detailed suggestions for simulation scenarios:

ll 115-117: how does that compare to the model presented here?

l 260: please specify how the R0=2.5 is derived from the estimates in [22] (which was 2.29 and 2.92, with and without rate heterogeneity among subepidemics resp).

ll 282ff & Fig 3: the upward vs downward bias changes at CV~SQRT(2) for the 3 parameters R0, mu, gamma. and the same is seen in Fig 5. I doubt this is a coincidence, but that it has to do with the characteristics of the gamma function employed to model heterogeneity (if I understand correctly CV_lambda is the inverse squareroot of the shape parameter, is that right?). the authors should explore this and discuss how this may be used to assess in which direction they expect bias to occur. 

simulations:

- what if sampling is delayed?  - this would probably make the inference harder but would be more realistic. I suggest to add a simulation scenario to test that.

real data analyses (page 10 / Fig 7): 

- please provide more information on the real datasets, at least the sampling period and number of samples

- please perform additional simulations that test significance of the real dataset analyses, i.e. (i) a slow outbreak with high heterogeneity and (ii) a fast outbreak with low heterogeneity, and compare the results to each other and the real data analyses resp. 

- why was a coalescent skyline used in the BEAST analyses? the tree prior may impact the tree characteristics esp. in the MSM_B dataset, which appears quite small, such that the coalescent skyline may be "overfitted"

- I would guess that the low gamma estimate in IDU_B is an artefact of the root of the tree ranging much further back than the oldest sample. this translates to a very long infectious period for the first cases, but is more likely a bias in the sampling process. can the model account for that by letting the sampling process start later? 

phylogeny guided contact tracing:

- how robust is the NCE to biased sampling processes? parts of the tree that are sampled more heavily will have higher NCEs.

- this needs to be tested using simulations - applying heterogenous sampling schemes

- with large enough datasets a more complex birth-death model may be used to infer such differences, which could then be used to adjust the NCE accordingly.

ll 452-453 "the effect of within host diversity is to make short branches only slightly longer whereas longer branches are increased more due to within host diversity"  - why is that?

Fig 8 & ll 491ff are 20% and 50% realistic proportions for contact tracing? please provide some references for that

-  Fig 8 would be easier to read if the authors used a colour gradient for the fraction contact traced 

ll 509-510: this has only been tested for very specific cases. please phrase this specifically referring to what you've tested, it's an unjustified generalisation

Discussion: there may also be heterogeneity through time, not only among individuals, particularly regarding the time to diagnosis

Parts of the supplement are confusing, e.g. the last sentence on page 3, most of Section 4.2 etc.

Reviewer #3: This manuscript introduces a new method to estimate transmission heterogeneity from viral sequence data. It is an important question and a successful such method is appropriate for publication in PLOS CB.

I have some concerns about the method and its assumptions.

Line 72 I don’t understand why you can say that the ratio of sequenced diagnosed over all infected (c /(a+b+c)) is the sampling ratio (c/(b+c)) unless a=0. Please rephrase, as this obviously isn’t what you mean.

Line 86 as you later clarify, you still can’t say G=T even when there is no in-host diversity, because T needs information about which host internal nodes are in (ie direction of transmission).

A birth-death model with constant rates (even rates that vary between individuals) is a very poor match for the within-host natural history of HIV infection, which has an early high-transmission phase, followed by a long chronic infection phase, and then followed by progression to AIDS (in the absence of treatment). With treatment, the assumption that transmission ceases is good in some settings (with great treatment adherence and little resistance, for example), and poorer in others. But in pretty much all settings, assuming a constant infectivity for 2.5 years is not a great model; this needs to be discussed.

The simulation study could explore model mis-specification in this direction.

Line 140 surely the likelihood is proportional to, not equal to, as k and theta have moved across the conditioning.

Does * mean multiplication in Eq (1), not convolution?

What is gamma^1_DG? - Ah, in line 143 you state that the indicator is for whether the infectious period ended before the observation period; surely the probability of this depends on when the infectious period started.

Pairwise transmission models have a long history and are well-liked in the community, but I have always wondered what the independence assumption means. Clearly if some individual A is in the dataset, then someone infected A (or A is the index case). If J infected A, then B did not infect A, nor C, D, …, because A can only have one infector. When we assume independence of all the transmission events, we cannot take these constraints into account.

Line 153 i don’t know what $(d_i, x_i)_{i=1}^n G$ is.

Line 163 - why would eq (1) be valid for a partially reconstructed transmission history? Or, what does that validity mean, given that the partially sampled history does not contain the information required to use (1) as it does not contain d_i and x_i (since it’s partially sampled)?

Line 183 - what does “each slice consists of branches having the same maximum number of branches to the tips” mean?

I like the observation in line 186 that the sum of the infectious periods has to add up to the total branch length; this applies to other models relating transmission to inference (phybreak, beastlier etc also) as far as I can tell.

Please state exactly what the CA algorithm does - alternating between relabelling and re-estimating sounds good but please give more details; pseudo-code.

It’s the section on analysis of the sampled genealogy where I really don’t follow and begin to doubt the modeling. First - is $t$ counted from t=0 at the root of the tree, or back into the past from the tips (and if so, which one)?

With no in-host diversity, are you assuming that the external branches represent time from infection to sampling? Why use L_p in the way that you do? Or do you mean the heights of the tips, in line 196?

You split the tree into the part before time t-Lp. There you estimate the sampling ratio; then you apply the CA algorithm.; then you correct the parameter estimates for Lp, and then “the results under different values of p are averaged” -- again, please clarify; give pseudo-code if possible.

I really don’t see why the sampling ratio of the G(s) (t-Lp) is rho_SD *p . You have *set* p, so p could be whatever you choose. How does Lp provide an estimation about the p percentile for the distribution of infectious periods, and how does this mean that you know the sampling ratio of G(s)(t-Lp)?

Also, if the sequencing ratio (and sampling fraction) is fixed over time, then you know from the times of sequenced cases quite a lot about the system.

Line 220 what is “upscaled” here?

Line 244 why should taxa with higher NCE values be more likely to be phylogenetically linked to unsampled highly infectious individuals? Does this happen in your simulations? I might have thought that in densely-branching parts of the phylogeny, sampling is likely to be more dense, not less.

R0=2.5 seems a high estimate for HIV in Switzerland; most estimates for developed countries without generalised epidemics are less than 1.

How robust is the method to model mis-specification? As above I would be most concerned about the non-constant transmission rate.

I think the main text should state more about the simulation method, as it is inconvenient to have to go and dig it out of a supplement; in particular, the fact that you simulate trees rather than sequences is quite important as it means that you cannot see how robust the method is to tree uncertainty; given that you required timed trees, there is also the potential for the nuances of timed tree reconstruction to contribute to bias or noise in the estimates.

Similarly, the main text should have information about how the within-host diversity was simulated (perhaps with a figure of a tree from such a simulation).

There are various plots with “coefficient of variation” as the title and throughout there are plot titles but no y axis labels, which left me confused. Fig 6 - CV_lambda is on the x axis; what is y. I can’t tell from the caption what the vertical axis is showing. In Fig 3, CV_\\lambda is defined as “levels of heterogeneity” - what is it, in terms of the parameters of the simulation model?

Figs 9, 10 what is on the y axis? The title “comparison of relative effect” doesn’t tell us much

Supplement S2

Why is the adjusted transmissibility just lambda_i/rho ?

**Have all data underlying the figures and results presented in the manuscript been provided?**

Reviewer #1: None

Reviewer #2: No: The authors state that "The alignments used for this study are available from the authors upon request"

 - I believe the code, analysis and alignment files should be published with the study, or a justification otherwise is necessary.

Reviewer #3: Yes

PLOS authors have the option to publish the peer review history of their article (what does this mean?). If published, this will include your full peer review and any attached files.

Reviewer #1: Yes: Art Poon

Reviewer #2: Yes: Denise Kühnert

Reviewer #3: No

---

## [Decision Letter · Decision Letter 1]

27 May 2020

Dear Dr. zhang,

Thank you very much for submitting your manuscript "Inferring transmission heterogeneity using virus genealogies: estimation and targeted prevention" for consideration at PLOS Computational Biology.

As with all papers reviewed by the journal, your manuscript was reviewed by members of the editorial board and by several independent reviewers. In light of the reviews (below this email), we would like to invite the resubmission of a significantly-revised version that takes into account the reviewers' comments.

We cannot make any decision about publication until we have seen the revised manuscript and your response to the reviewers' comments. Your revised manuscript is also likely to be sent to reviewers for further evaluation.

Sincerely,

Roger Dimitri Kouyos

Associate Editor

PLOS Computational Biology

Virginia Pitzer

Deputy Editor

PLOS Computational Biology

Reviewer's Responses to Questions

**Comments to the Authors:**

Reviewer #1: I thank the authors for their work in revising the manuscript and providing some documentation for their source code on the GitHub repository. There are a number of remaining issues that need to be corrected before this manuscript can be considered for publication.

Introduction, lines 24-34: The authors’ characterization of the multi-state birth-death (MSBD) model is incorrect. The MSBD model does not require the positions and times of state changes in the phylogeny to be known in advance — instead, the optimization algorithm adds and removes state changes in the phylogeny.

Results, lines 364-384: Your comparison of the uncorrelated model to the MMPP model is problematic because you have simulated transmission rates as being drawn individually (in an uncorrelated fashion) from a gamma distribution. This violates a central assumption of the MMPP, which is premised on the existence of correlated rates (which is how clusters are identified). It is therefore not at all surprising that your model performs well on data simulated under the same model, and MMPP performs poorly when its core assumption is broken. The honest approach would be to evaluate the methods under both uncorrelated and correlated scenarios, recognizing that MMPP should be unable to detect variation in transmission rates in the uncorrelated scenario. In addition, MMPP does not attempt to estimate R0 — it is misleading to present results (Figure S9) that indicate otherwise.

Introduction, lines 45-52: The addition of this text helps clarify that the authors’ model assumes that variation in transmission rates are uncorrelated (independent draws from some parametric distribution). However, the rationale provided by the authors — “varying degrees of social activity” — implies auto-correlation in transmission rates, since risk behaviours exhibit a high degree of homophily. Varying viral loads is time-heterogeneous within individuals, whereas your model assumes rates are time-invariant at the level of individuals. Thus, the rationale for your model remains unclear.

Methods, lines 261-280: The addition of this section helps clarify the core assumptions of the authors’ model. Another way to put this is that the authors assume that, in the setting of high variation in transmission rates, all “superspreader” individuals have been sampled and diagnosed. The authors need to reconcile this assumption with another central assumption of their model, that the sampling and diagnosis of cases from the epidemic is occurring at an early exponential phase of the epidemic. (This would involve a tremendous sampling effort because most individuals will be uninfected when sampled.)

For equation (9), I can see that the authors intend to scale the constant transmission rate for unsampled individuals from \\mu_lambda^0 at CV=0 to \\gamma as CV\\leftarrow\\infty. I cannot reconcile this with the statement at the top of this paragraph that \\lambda _{us} = \\mu_lambda^0 / \\rho. It is also important to clarify for the reader that there is no variation in transmission rates among unsampled individuals under any level of CV in this model.

**Have all data underlying the figures and results presented in the manuscript been provided?**

Reviewer #1: None

PLOS authors have the option to publish the peer review history of their article (what does this mean?). If published, this will include your full peer review and any attached files.

Reviewer #1: No
---

## [Decision Letter · Decision Letter 2]

2 Jul 2020

Dear Dr. zhang,

We are pleased to inform you that your manuscript 'Inferring transmission heterogeneity using virus genealogies: estimation and targeted prevention' has been provisionally accepted for publication in PLOS Computational Biology.

Before your manuscript can be formally accepted you will need to complete some formatting changes, which you will receive in a follow up email. A member of our team will be in touch with a set of requests. In addition, the reviewer pointed out a few typographical errors that you may want to address.

Best regards,

Roger Dimitri Kouyos

Associate Editor

PLOS Computational Biology

Virginia Pitzer

Deputy Editor

PLOS Computational Biology

Reviewer's Responses to Questions

**Comments to the Authors:**

Reviewer #1: Thank you for addressing my comments with this revised manuscript, which is substantially improved.

Regarding the authors' assertion that "When A was unsampled while B and C were sampled, the coalescent event of B and C in the sampled tree (with tips B and C) is coincide with that of A and B in the full tree (with tips A~C)." -- I think this requires a strong assumption about the coalescence of lineages within hosts. While I agree that lineages are more likely to be traced back to a "superspreader" host, there is no guarantee that the coalescent events involving different sampled hosts will coincide (especially if the transmission bottleneck is incomplete). This is a minor point, however.

I only have some typographical or stylistic errors to report that I would leave to the authors' discretion to amend.

line 396: "it is clearly that" should be "it is clear that"

line 400: dropped backslash resulted in exposed "mu_\\lambda".

lines 396-403: it would be helpful to note the number of replicate simulations evaluated here

Figure S9: "Virus genealogies were simulated under two scenarios: with autocorrelation in transmission rates (A and B) and without autocorrelation (C and D)." -- according to the figure titles, this is backwards; A and B are *without* autocorrelation.

**Have all data underlying the figures and results presented in the manuscript been provided?**

Reviewer #1: None

PLOS authors have the option to publish the peer review history of their article (what does this mean?). If published, this will include your full peer review and any attached files.

Reviewer #1: No

---

## [Editor Report · Acceptance letter]

26 Aug 2020

PCOMPBIOL-D-19-01410R2 

Inferring transmission heterogeneity using virus genealogies: estimation and targeted prevention

Dear Dr zhang,

I am pleased to inform you that your manuscript has been formally accepted for publication in PLOS Computational Biology. Your manuscript is now with our production department and you will be notified of the publication date in due course.

With kind regards,

Matt Lyles
